

# A Probabilistic Approach to Wildfire Spread Prediction Using a Denoising Diffusion Surrogate Model

Wenbo Yu[1,2], Anirbit Ghosh[3], Tobias Sebastian Finn[1], Rossella Arcucci[2,4], Marc Bocquet[1], and Sibo Cheng[1]

[1]CEREA, ENPC, EDF R&D, Institut Polytechnique de Paris, Île-de-France, France
[2]Department of Earth Science & Engineering, Imperial College London, London, UK
[3]Department of Computing, Imperial College London, London, UK
[4]Data Science Institute, Imperial College London, London, UK

**Correspondence:** Sibo Cheng (sibo.cheng@enpc.fr)

**Abstract.** We propose a stochastic framework for wildfire spread prediction using deep generative diffusion models with ensemble sampling. In contrast to traditional deterministic approaches that struggle to capture the inherent uncertainty and variability of wildfire dynamics, our method generates probabilistic forecasts by sampling multiple plausible future scenarios conditioned on the same initial state. As a proof-of-concept, the model is trained on synthetic wildfire data generated by a

probabilistic cellular automata-based simulator, which integrates realistic environmental features such as canopy cover, vegetation density, and terrain slope, and is grounded in historical fire events including the Chimney and Ferguson fires. To assess predictive performance and uncertainty modelling, we compare two surrogate models with identical network architecture: one trained via conventional supervised regression, and the other using a conditional diffusion framework with ensemble sampling. In the diffusion-based emulator, multiple inference passes are performed for the same input state by resampling the initial latent variable, allowing the model to capture a distribution of possible outcomes. Both models are evaluated on an independent

ensemble testing dataset, ensuring robustness and fair comparison under unseen wildfire scenarios. Experimental results show that the diffusion model significantly outperforms its deterministic counterpart across various metrics. At a training size of 900, the diffusion model outperforms the deterministic baseline by a substantial margin. Averaged across the Chimney fire and Ferguson fire datasets, the diffusion model achieves a 67.6% reduction in mean squared error (MSE), a 5.4% improvement in

structural similarity index (SSIM), and a 69.7% reduction in Fréchet Inception Distance (FID). These findings demonstrate that diffusion-based ensemble modelling provides a more flexible and effective approach for wildfire forecasting. By capturing the distributional characteristics of future fire states, our framework supports the generation of fire susceptibility maps that offer actionable insights for risk assessment and resource planning in fire-prone environments.

## 1 Introduction

Climate change has amplified extreme weather events, driving an increase in the frequency and scale of wildfires worldwide. These fires have devastating impacts on infrastructure, human safety, ecosystems, and the environment (Gajendiran et al., 2024). Each year, millions of hectares of forest are destroyed, resulting in the loss of wildlife habitats and plant communities,



heightened greenhouse gas emissions, and severe economic and human casualties (Sun et al., 2024; Pelletier et al., 2023). To address these challenges, physics-driven probabilistic models such as Cellular Automata (Freire and DaCamara, 2019)

and Minimum Travel Time (Finney, 2002) have been developed to simulate the spread of wildfires under real geographic conditions. Despite their ability to model fire dissemination patterns, these approaches face significant limitations in speed and computational efficiency due to their dependence on extensive geophysical and climate data. Furthermore, physical models often lack robustness to environmental variations due to limited explanations of combustion mechanisms (Jiang et al., 2023) and demand substantial computational resources to solve complex conservation equations and require specialized expertise,

making them challenging to design and implement (Makhaba and Winberg, 2022; Jiang et al., 2023).

In recent years, Machine learning (ML) and Deep learning (DL) methods have been extensively employed to tackle the issue of wildfire detection and wildfire spread prediction, with recent reviews in Jain et al. (2020); Xu et al. (2024). A lot of them employ DL models (in particular, convolutional and recurrent neural networks) to emulate existing physics-based fire spread models and enhance their computational efficiency (Marjani et al., 2023; Singh et al., 2023; Cheng et al., 2022).

Advanced architectures such as Transformer and U-Net, often enhanced with attention mechanisms, have emerged as key approaches in wildfire spread prediction research (Shah and Pantoja, 2023; Chen et al., 2024b). Notable examples include the FU-NetCastV2 model (Khennou and Akhloufi, 2023) and the Attention U2-Net (Shah and Pantoja, 2023), both leveraging U-Net architectures to achieve impressive prediction accuracy. Similarly, the AutoST-Net (Chen et al., 2024b), integrates transformer mechanisms and 3DCNNs to effectively capture the spatiotemporal dynamics of wildfire spread, outperforming

CNN-LSTM model (Bhowmik et al., 2023) with a 6.29% increase in F1-score on a wildfire dataset constructed using Google Earth Engine (GEE) (Gorelick et al., 2017) and Himawari-8 (Bessho et al., 2016).

Despite these advancements, current ML approaches to wildfire spread prediction predominantly rely on deterministic models. This reliance significantly limits their capacity to account for the stochastic nature of wildfire dynamics, which are profoundly influenced by complex and variable factors such as wind speed, canopy density, and topographical variations. Such

models struggle to reflect the inherent uncertainty and variability of natural systems, which is particularly problematic for phenomena like wildfire spread, where minor changes in environmental conditions can result in vastly different outcomes (Holsinger et al., 2016; Dahan et al., 2024). As a result, deterministic ML techniques fail to capture this stochastic behaviour of wildfire propagation.

To address the limitations of deterministic approaches, researchers have increasingly turned to ensemble methods and

stochastic frameworks to better capture the uncertainty and variability of wildfire dynamics. Ensemble-based simulation systems, such as the method proposed by Finney et al. (2011), utilise synthetic weather sequences to perform thousands of fire growth simulations, generating spatial probability fields that reflect potential fire behaviours under varied conditions. These methods provide a probabilistic perspective, which is particularly valuable for long-term wildfire risk assessments but are constrained by computational demands (Finney et al., 2011). More recent advancements have leveraged machine learning-based

ensemble models to enhance computational efficiency and uncertainty quantification. For instance, the SMLFire1.0 framework introduced by Buch et al. (2023) employs stochastic machine learning techniques to model wildfire frequency and the area burned across diverse ecological regions, offering robust correlations with observed data and highlighting key fire drivers such





as vapor pressure deficit and dead fuel moisture (Buch et al., 2023). In a related domain, deep learning has shown promise in improving stochastic processes in climate models, where traditional deterministic methods often fall short. For example, Behrens et al. developed stochastic parameterizations for subgrid processes in Earth System Models (ESMs) using deep learning, illustrating how ensemble methods improve the representation of convective processes and enhance uncertainty quantification. Similarly, Bjånes et al. (2021) created a deep learning ensemble model to integrate static and dynamic variables for producing wildfire susceptibility maps, achieving high predictive performance and offering actionable insights for fire prevention and resource allocation. These advances underscore a broader shift towards incorporating stochastic and ensemble-based approaches, which bridge the gap between deterministic modelling and the complex, probabilistic nature of wildfire dynamics.

Building on these advancements, this study investigates the use of diffusion models within an ensemble prediction framework to address the inherent limitations of deterministic methods in predicting wildfire spread patterns. Diffusion models (Ho et al., 2020) have demonstrated significant efficacy across a wide range of disciplines, including image and audio generation, natural language processing, and life sciences (Chen et al., 2024a). Its advantages over conventional generative methods, such as Variational Autoencoders (VAEs) and Generative Adversarial Networks (GANs), have been demonstrated in numerous studies (Dhariwal and Nichol, 2021; Vivekananthan, 2024), particularly in addressing challenges like mode collapse and blurred outputs.

Recent applications of diffusion models to complex, unstructured data have further highlighted their potential in dynamic system modelling. For instance. Price et al. (2024) introduced GenCast, a diffusion-based ensemble forecasting model that surpasses state-of-the-art numerical weather prediction systems in skill and efficiency for medium-range global weather forecasts. Similarly, Finn et al. (2024) demonstrated the use of generative diffusion models for regional surrogate modelling of sea-ice dynamics, showing that these models outperform traditional approaches in accuracy while being orders of magnitude faster; Nath et al. (2024) utilised cascaded diffusion models for forecasting precipitation patterns and cyclone trajectories through the integration of satellite imagery and atmospheric datasets; Leinonen et al. (2023) and Gao et al. (2023) proposed latent diffusion models for near-term precipitation forecasting, demonstrating the ability to accurately capture forecast uncertainty while producing high-quality and diverse outputs. These studies exemplify the versatility of diffusion models across geoscientific domains, where they effectively manage high-dimensional datasets, capture complex spatiotemporal dynamics, and offer robust probabilistic predictions. Thus, diffusion models stand out for their probabilistic generative framework, which allows them to model a range of potential wildfire propagation scenarios rather than producing a single deterministic forecast. The diffusion model in this study serves as an emulator (also known as a surrogate model), learning to approximate the behaviour of a probabilistic Cellular Automata (CA) model Alexandridis et al. (2008) that simulates wildfire spread under realistic environmental conditions. Since the wildfire spread is simulated using a Cellular Automata framework, which operates in a discrete state space, the diffusion model is trained to emulate this process by predicting binary outcomes representing burned (1) or unburned (0) states at each grid cell. Recent advances in discrete diffusion models, such as Structured Denoising Diffusion Models (D3PMs) Austin et al. (2021) and Bit Diffusion Chen et al. (2023), provide theoretical justification for adapting diffusion models to binary prediction tasks. These methods demonstrate that diffusion-based frameworks can effectively model structured discrete variables, concentrating probability mass on physically meaningful states, which is particularly relevant for



modelling wildfire propagation. To the best of the authors' knowledge, no previous studies have explored the use of diffusion models as emulators for predicting wildfire spread using real-world ecoregion data.

The primary objective of this research is to develop a diffusion-based emulator guided by initial wildfire conditions and capable of accurately simulating potential fire spread scenarios. The proposed model will leverage terrestrial image sequences of burned area evolution over time as training data. A probabilistic Cellular automata (CA) model, as proposed by Alexandridis et al. (2008), will be used to generate these sequences based on the real-world wildfire events. Additionally, this study makes use of an ensemble sampling method, as similarly used in other diffusion models for geophysical forecasting. This method involves performing multiple inference passes, as shown in Figure 1, each generating a potential outcome, and then averaging these outcomes to produce an ensemble prediction. We will show that this ensemble approach effectively captures the underlying uncertainty and variability of wildfire spread, providing a probabilistic forecast that is crucial for informed wildfire management and decision-making.

By leveraging the strengths of diffusion models as emulators, this study seeks to address the limitations of traditional deterministic wildfire forecasting methods, providing a more flexible and probabilistic framework for fire prediction. In summary, the main contributions of this work are as follows:

- We introduce an ensemble sampling method for wildfire spread prediction, which leverages a diffusion-based generative framework and demonstrates superior accuracy and robustness compared to deterministic state-of-the-art models.

- By modelling a probability density of future fire states, our approach more effectively captures the inherent uncertainty in wildfire dynamics than traditional deterministic methods, enabling the creation of fire susceptibility maps that represent the expected likelihood of fire spread and offer valuable insights for risk assessment and management.

- The method is evaluated on data produced by a probabilistic Cellular Automata-based emulator that integrates realistic environmental features, including canopy cover, canopy density, and landscape slope, ensuring robust and credible performance across diverse wildfire scenarios.

The rest of this paper is organized as follows: Section 2 presents the data preparation process, including the study area, and details about the burned area dataset, including both the training and testing datasets. Section 3 describes the methodology, detailing the diffusion model for wildfire prediction, the forward and backward processes, model training, inference, and the model architecture. Section 4 discusses the experimental design, followed by an analysis of the results, including overall performance and the impact of diffusion sampling times on performance metrics. Finally, Section 5 concludes on the findings, limitations, and potential future directions.



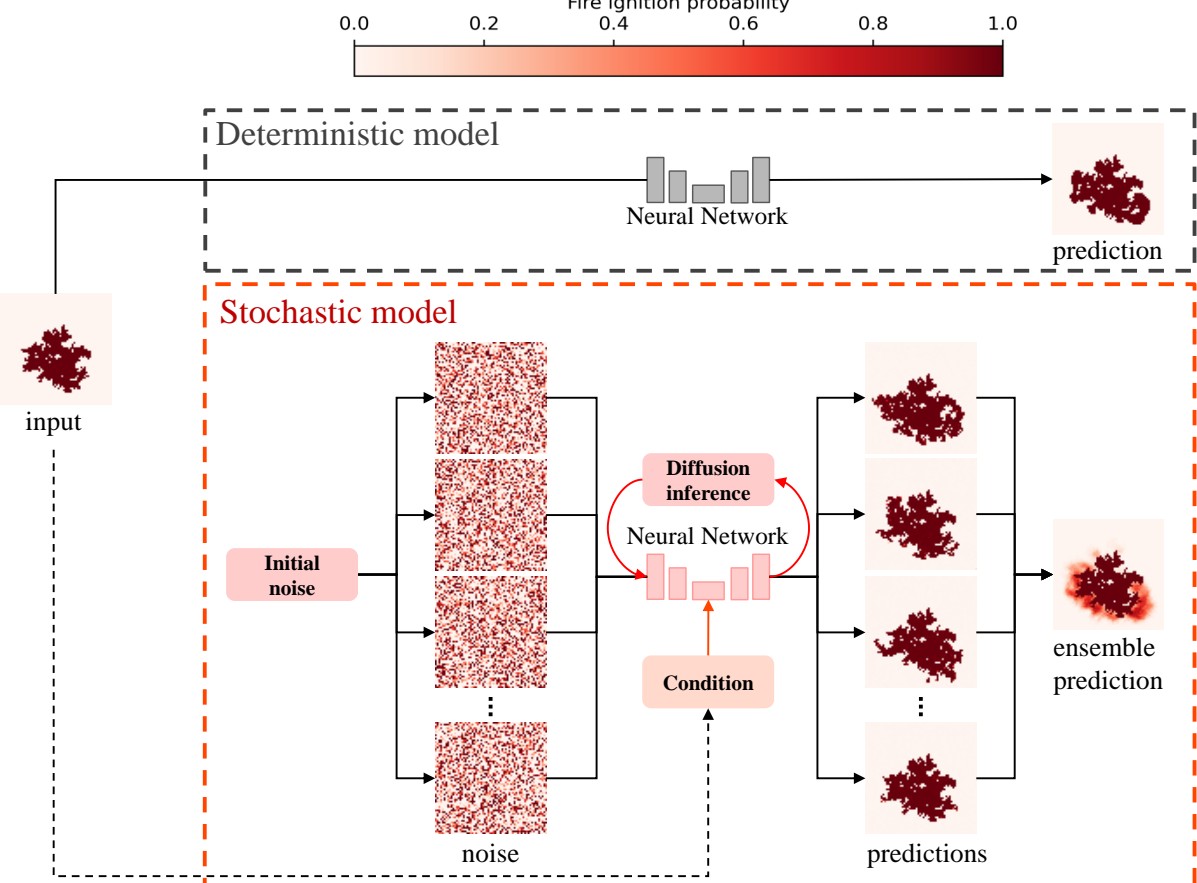

**Figure 1.** Deterministic and stochastic models.

## 2 Data

In this section, we describe the data used to train the predictive diffusion model, including details about the study area and its geological characteristics, followed by a description of the CA model employed for generating experiment data.

### 2.1 Data preparation

#### 2.1.1 Study area

This case study evaluates the performance of wildfire prediction models using data collected from the 2016 Chimney fire (Walpole et al., 2020) and the 2018 Ferguson fire (Wang et al., 2021) in California (see Table 1 for details). For instance, in this proof-of-concept study, two distinct diffusion models are trained for each ecoregion. The canopy density in the area affected by the Chimney fire was higher than that of the Ferguson fire, leading to a considerably faster rate of spread. Consequently, these



two fires exhibit contrasted behaviours, providing a valuable basis for assessing the effectiveness and robustness of stochastic
modelling approaches in wildfire prediction.

| Fire | Latitude | Longitude | Area | Duration | Start | Wind |
|---|---|---|---|---|---|---|
| Chimney | 37.6230 | -119.8247 | $\approx 246$ km$^2$ | 23 days | 13 August 2016 | 23.56 mph |
| Ferguson | 35.7386 | -121.0743 | $\approx 185$ km$^2$ | 36 days | 13 July 2018 | 18.54 mph |

**Table 1.** Information of the study areas used in this research, including details on fire incidents, their geographic coordinates, affected area,
duration, start date, and average wind speed.

### 2.1.2 Stochastic cellular automata wildfire simulator

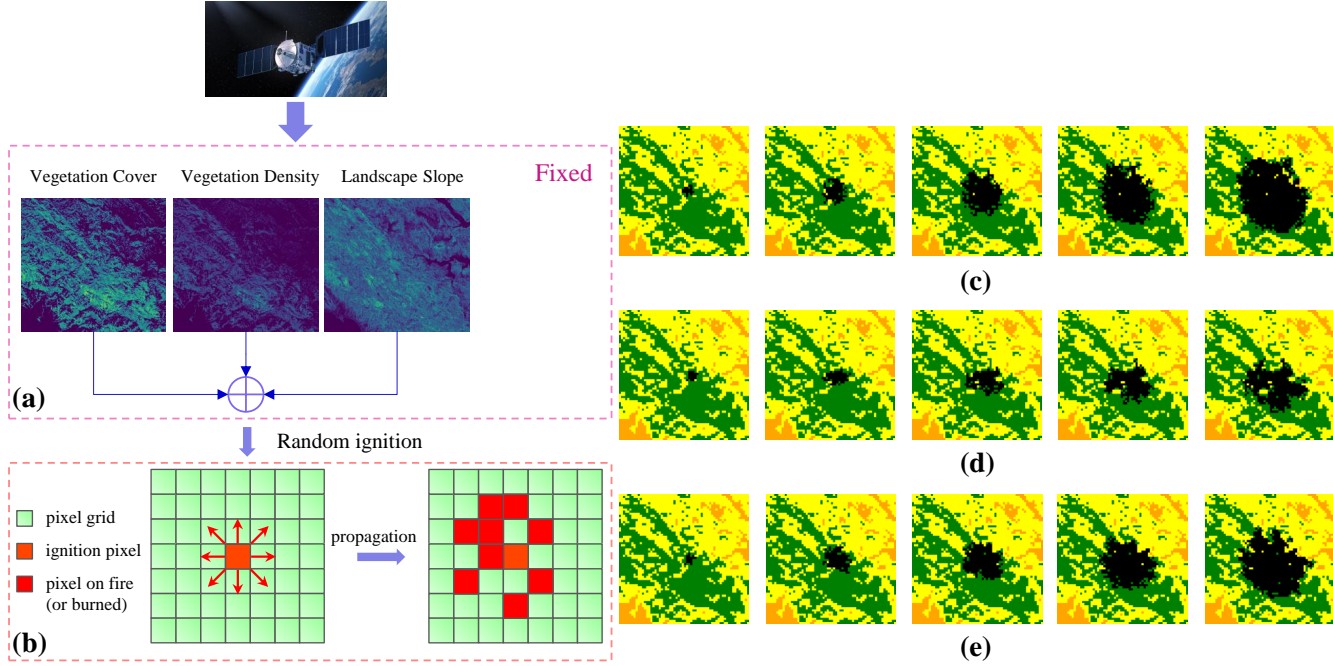

**Figure 2.** (a) Data collection, including canopy density, canopy cover, landscape slope and local wind speed corresponding to the forest area
affected by the Chimney fire, California, in 2016; (b) Possible directions considered for each cell when simulating fire propagation using
Cellular automata (CA). (c-e) CA simulated wildfire spread samples from random ignition points at intervals of 20 hours.

In this study, the experimental data used to train and evaluate the diffusion model were generated using a CA model. This
CA model is based on the framework developed by Alexandridis et al. (2008), which was originally designed to simulate the
dynamics of wildfire spread in mountainous landscapes. The original study demonstrated the effectiveness of the CA model



in replicating the spread of the 1990 wildfire on Spetses Island, proving its potential as a valuable tool for simulating wildfire spread scenarios (Alexandridis et al., 2008).

This method entails the partitioning of the forest area into a grid of square units encoded in state matrices with two-dimensional coordinates, each of which is susceptible to eight potential directions of fire spread Alexandridis et al. (2008).

As shown in Figure 2.b, the model considers all eight cardinal and ordinal directions. The cells in the grid are discretised into four possible states: i) unburnable cells, ii) cells that have not been ignited, iii) burning cells, iv) cells that has been burned down (Alexandridis et al., 2008). The probability of the fire igniting the adjacent unit on the next time step, denoted $p_{\mathrm{burn}}$, can be calculated by

$$p_{\mathrm{burn}} = p_{\mathrm{h}}(1 + p_{\mathrm{veg}})(1 + p_{\mathrm{den}})p_{\mathrm{wind}}p_{\mathrm{slope}}, \tag{1}$$

where $p_{\mathrm{h}}$ represents a standard burning probability, $p_{\mathrm{veg}}$, $p_{\mathrm{den}}$, $p_{\mathrm{wind}}$ and $p_{\mathrm{slope}}$ indicate the local canopy cover, canopy density, wind speed and landscape slope (Alexandridis et al., 2008). The implementation of the Cellular automata (CA) model and its parameter settings in this study are based on prior work by Cheng et al. (2022).

The geophysical and environmental data required for wildfire simulation were derived from remote sensing data, primarily obtained from the MODIS satellite (Giglio et al., 2016). These data, accessible through the Interagency Fuel Treatment Decision Support System (IFTDSS) (Drury et al., 2016), provide critical information on active fire locations, canopy cover, land surface conditions and wind speed as an atmospheric forcing. The burn probability model ($p_{\mathrm{burn}}$) within the CA simulator relies on these geophysical inputs to estimate fire spread dynamics. An instance of the vegetation data and a simulated fire propagation is shown in Figure 2.a for the Chimney fire ecoregion. The atmospheric forcing in this proof-of-concept study is set to constant values, following those used in the previous study by Cheng et al. (2022).

## 2.2 Burned area dataset

For the purposes of this study, the state classification was simplified by merging the original four categories into two: 0 – unburnt cells, combining unburnable and unignited cells; 1 – burnt cells, combining currently burning and already burnt cells.

The CA simulation was conducted at the state level, with each state saved as a grayscale image snapshot, wherein each cell in the grid was represented by a single pixel in the image. The binary images effectively captured the spatial distribution of the fire spread at each time step, thereby providing a visual representation of the wildfire dynamics over time, which represents the probabilistic fire spread at the next time step.





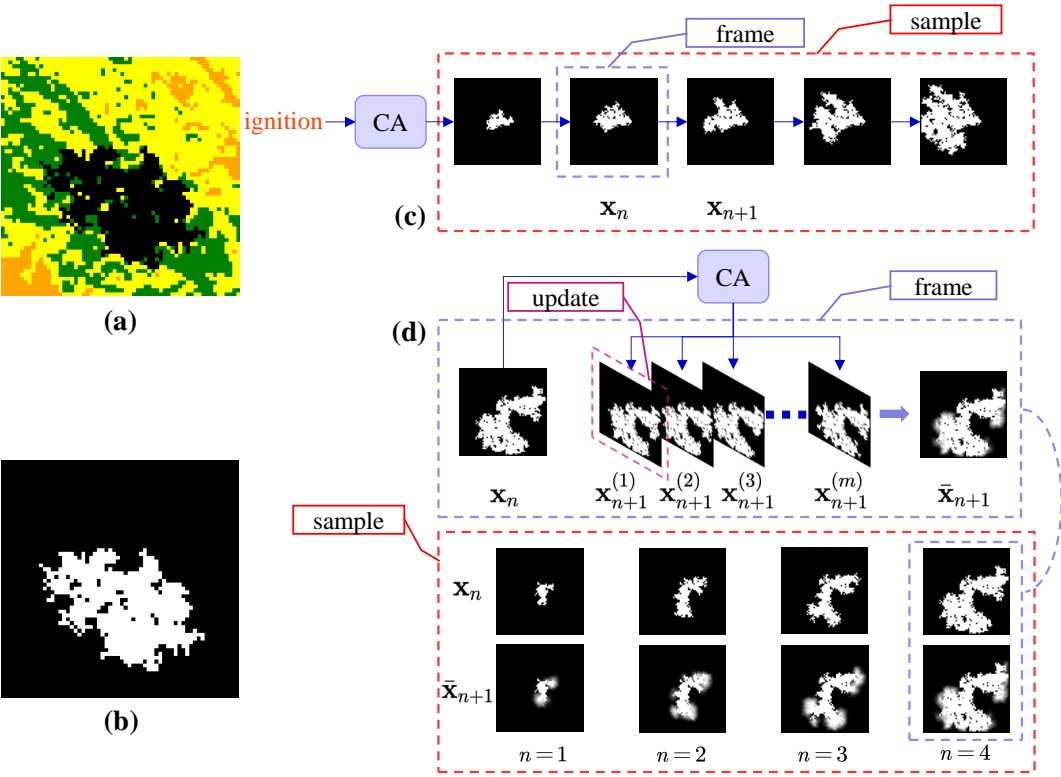

**Figure 3.** (a) A snapshot (frame) of a sample of wildfire spread simulated with CA. (b) Grayscaled snapshot. (c) A complete wildfire spread simulation sequence is recorded as a sample where each snapshot is a frame. (d) An example illustrating the creation of the ensemble testing dataset. Each sample in the dataset is generated with a different initial ignition position and consists of a sequence of input-target frame pairs. The CA model is executed multiple times starting from a single frame $\mathbf{x}_n$, producing multiple simulated versions of the next frame $\mathbf{x}_{n+1}$. These simulated frames are then averaged to produce an ensemble simulation $\bar{\mathbf{x}}_{n+1}$, representing the probability of fire spread at the next time step. Each pair of frames $(\mathbf{x}_n, \bar{\mathbf{x}}_{n+1})$ forms an input-target pair, providing data for evaluating the model's ability to predict the probabilistic transition between consecutive wildfire states.

### 2.2.1 Training dataset

The training dataset is a binary image dataset generated by running the CA simulator multiple times with varying ignition positions. Each simulation produces a sequence of wildfire spread states, capturing the temporal evolution of a single fire event from ignition to complete burnout across the entire grid. Each data sample represents the wildfire spread trajectory simulated from a specific ignition position, comprising a sequence of binary snapshots that capture the evolution of the wildfire state over time.

As illustrated in Figure 3.c, each simulation of a fire event is discretised into snapshots taken at 2-hour intervals. A complete simulation consists of 51 frames, denoted as $\{\mathbf{s}_0, \mathbf{s}_1, \cdots, \mathbf{s}_{50}\}$, where $\mathbf{s}_0$ represents the initial state of the wildfire at ignition,





and $s_i (i > 0)$ denotes the wildfire state at the $i$-th 2-hour time step. To generate the dataset for model training, frames are subsampled from each simulation at intervals of 10 time steps (i.e., every 20 hours), resulting in six frames per wildfire event sample: $\{\mathbf{s}_{10}, \mathbf{s}_{20}, \mathbf{s}_{30}, \mathbf{s}_{40}, \mathbf{s}_{50}\}$. From these frames, six input-target pairs $(\mathbf{x}_n, \mathbf{x}_{n+1})$ are created, where $\mathbf{x}_n = \mathbf{s}_{10 \times n}$ denotes the corresponding target frame at the subsequent sampled step. This dataset is used as the training set for the model, providing necessary data for the model to learn the stochastic processes underlying wildfire propagation over time.

## 2.2.2 Ensemble testing dataset

An ensemble testing dataset was constructed specifically for evaluation purposes to assess whether the generative predictive model accurately represents the probability density distribution of wildfire spread. As depicted in Figure 3.d, each sample in the ensemble dataset consists of a sequence of frame pairs, where each pair comprises a wildfire state at a given time step and the corresponding ensemble-predicted next state.

For instance, given a wildfire spread trajectory represented by a sequence of fire frames $\{\mathbf{x}_1, \mathbf{x}_2, \ldots, \mathbf{x}_n \ldots\}$, the input-target pairs in the ensemble dataset are defined as $\{(\mathbf{x}_1, \bar{\mathbf{x}}_2), (\mathbf{x}_2, \bar{\mathbf{x}}_3), \ldots, (\mathbf{x}_n, \bar{\mathbf{x}}_{n+1}) \ldots\}$, where each $\bar{\mathbf{x}}_{n+1}$ is the ensemble next frame corresponding to $\mathbf{x}_n$. The start frame $\mathbf{x}_1$ in each sample is generated from independently simulated wildfire trajectory, each initiated from a new randomly chosen ignition position. These trajectories are entirely separate from those in the training dataset, ensuring an unbiased evaluation of the model's predictive capabilities. Each wildfire state $\mathbf{x}_n$ at a specific time step

within its respective trajectory serves as the input condition for generating the corresponding ensemble-predicted next state $\bar{\mathbf{x}}_{n+1}$.

To generate the ensemble next frame $\bar{\mathbf{x}}_{n+1}$, the CA model is executed $m$ times from the same wildfire state $\mathbf{x}_n$, where $m$ is the ensemble size parameter determining the number of simulations performed for each target frame. Each simulation produces a potential outcome of wildfire spread at the subsequent time step, capturing the stochastic nature of fire propagation.

The resulting frames, denoted as $\{\mathbf{x}_{n+1}^{(1)}, \mathbf{x}_{n+1}^{(2)}, \cdots, \mathbf{x}_{n+1}^{(m)}\}$, represent diverse possible transitions of the wildfire state. These frames are then averaged to produce the ensemble next frame $\bar{\mathbf{x}}_{n+1}$, where each pixel value represents the probability of that cell being burnt, ranging from 0 (unburnt) to 1 (burnt). Figure 3.d illustrates this process, where multiple potential outcomes for $\mathbf{x}_{n+1}$ are generated through repeated CA simulations and subsequently averaged to construct $\bar{\mathbf{x}}_{n+1}$, providing a probabilistic representation of wildfire spread. Each pair $(\mathbf{x}_n, \bar{\mathbf{x}}_{n+1})$ forms an input-target pair in the ensemble testing dataset, with $\mathbf{x}_n$

serving as the input and $\bar{\mathbf{x}}_{n+1}$ as the probabilistic target.

The ensemble testing dataset serves as the evaluation set for the model, enabling a detailed comparison between the model's predictions and the ensemble target frames. This comparison allows for the assessment of the model's ability to accurately capture the expected spread of the wildfire, particularly in terms of its representation of the probability density distribution across possible outcomes.





## 3 Methodology

### 3.1 Diffusion model for wildfire prediction

Diffusion models represent a class of generative models that learn to iteratively produce new data samples from noise. During training, noise is progressively added and removed to known trainings data through a forward (diffusion) process and a backward (denoising) process. Training a neural network as denoiser allows us to denoise samples of pure noise into clean data, even when the forward process is unknown during prediction. By altering the initial noise, we can generate different data samples. This allows us to produce an ensemble of forecasts and, hence, the probability distribution of future wildfire spread that is not available in more commonly used deterministic models.

The goal of our diffusion model is to produce a prediction $\mathbf{x}_{n+1}$ based on the initial conditions $\mathbf{x}_n$. By taking these initial conditions as additional input to approximate the score function, the neural network is trained as conditional diffusion model. For training and prediction, the forward and backward process are discretized in pseudo timesteps $t \in [0, T]$. These pseudo timesteps are independent from the real-time progression of the wildfire, they rather specify where we are in the diffusion process. Depending on the pseudo timestep, the noised state $\mathbf{x}_{n+1}^t$ contains a progressively increasing portion of noise. Note for the ease of notation we drop the subscripted real-time index $n+1$ in the following description of the diffusion model.

The forward process between two consecutive timesteps $t-1$ and $t$ is characterized by the transition probability $q(\mathbf{x}^t \mid \mathbf{x}^{t-1})$ with an explicit Markovian assumption. At the end of this process, at time $t = T$, almost all data is replaced by noise such that $\mathbf{x}^T \approx \boldsymbol{\epsilon}$ with $\boldsymbol{\epsilon} \sim \mathcal{N}(\mathbf{0}, \mathbf{I})$ holds, as shown in the upper blue part of Figure 4.

The backward process makes use of the transition probability $p_\theta(\widehat{\mathbf{x}}^{t-1} \mid \widehat{\mathbf{x}}^t)$, which includes the approximated denoiser with its parameters $\boldsymbol{\theta}$. Starting from pure noise $\boldsymbol{\epsilon}$, this process results into the reconstructed state $\widehat{\mathbf{x}}^t$ at pseudo timestep $t$, as shown in the lower green part of Figure 4. This chain of reconstructed states leads then to the prediction $\widehat{\mathbf{x}}_{n+1}^0 = D(\boldsymbol{\epsilon}, \mathbf{x}_n)$, the final output of the diffusion model. As a consequence of our trainings dataset, the diffusion model will produce predictions with modes around 0 (unburnt) and around 1 (burned), even if the output is continuous (Chen et al., 2023). The output of the diffusion model will be practically like a binary response.

Since the reconstruction is driven by the initial noise and intermediate draws from the transition probability, the diffusion model can produce different predictions from the same initial conditions. By sampling $m$-times different noise samples, the diffusion model generates effectively an ensemble of predictions, denoted as set $(\widehat{\mathbf{x}}_{n+1}^{0,(1)}, \widehat{\mathbf{x}}_{n+1}^{0,(2)}, \ldots, \widehat{\mathbf{x}}_{n+1}^{0,(m)})$. Based on this ensemble, we can evaluate the probability that a cell is burned.





**Figure 4.** Diffusion model training process

In the context of wildfire prediction, the diffusion model is designed to estimate the potential spread of a fire by predicting the area burned over time. The input to the model is the current state of the wildfire, represented as a fire state frame $\mathbf{x}_n$, which contains spatial information about the fire's extent at a given time. This frame is encoded as a two-dimensional matrix, where each element indicates a specific location in the region of interest. As described in the Data preparation section, a value of 0 signifies that the location is unburnt, whereas a value of 1 denotes that the location is actively burning or has already burned.

The diffusion model typically has trainable parameters, denoted by $\theta$. These parameters govern the neural network, sometimes called the "noise predictor", which is the actual component trained during the training process. During the forward (diffusion) process, Gaussian noise is systematically added to the data in fixed increments according to a Markov chain, and thus no prediction is performed in this phase. After training, the same neural network is used for "inference" in the backward (denoising) process, which will be explained in the next section.





### 3.1.1 Forward (diffusion) process

The forward process of the diffusion model progressively adds Gaussian noise to a particular state, denoted by $\mathbf{x}_n^0$ or $\mathbf{x}^0$ (representing the initial, unaltered state before any noise addition), across a series of $T$ steps governed by an approximate

posterior distribution $q(\cdot)$. In this setting, the variable $t \in \{1, \ldots, T\}$ is a pseudo-timestep within the diffusion process, rather than a direct representation of the actual temporal index of the wildfire progression. Formally, owing to Markov assumptions in the transition densities, the forward process is expressed as

$$q(\mathbf{x}^{1:T} \mid \mathbf{x}^0) = \prod_{t=1}^{T} q(\mathbf{x}^t \mid \mathbf{x}^{t-1}), \tag{2}$$

where $\mathbf{x}^{1:T}$ is the sequence of intermediate frames $(\mathbf{x}^1, \mathbf{x}^2, \ldots, \mathbf{x}^T)$ obtained by successively adding noise over $T$ pseudo-

timesteps, and $q(\mathbf{x}^{1:T} \mid \mathbf{x}^0)$ is the joint distribution over all these frames, given the initial state $\mathbf{x}^0$ and leading to the final noisy frame $\mathbf{x}^T$. The transition probability between consecutive frames at steps $t-1$ and $t$ is chosen to be

$$q(\mathbf{x}^t \mid \mathbf{x}^{t-1}) = \mathcal{N}\left(\mathbf{x}^t; \sqrt{1-\beta^t}\,\mathbf{x}^{t-1}, \beta^t \mathbf{I}\right), \tag{3}$$

where $\mathcal{N}(\cdot)$ denotes a Gaussian distribution with $\beta^t$ as the variance of the noise added at step $t$, and $\mathbf{I}$ as the identity matrix, indicating isotropic noise. The values of $\beta^t$ are set according to a linear schedule increasing from a small initial value to a

maximum value at $T$, following the approach in Ho et al. (2020), ensuring a gradual and controlled diffusion process. As $t$ progresses from 1 to $T$, the state $\mathbf{x}^t$ becomes increasingly noisy.

### 3.1.2 Backward (denoising) process

The reverse process, also referred to as the denoising process, is the generative component of the diffusion model, which iteratively removes noise introduced during the forward process. In the context of wildfire prediction, this corresponds to the

inference phase, where the goal is to estimate the subsequent wildfire state $\mathbf{x}_{n+1}^0$ by refining a sequence of intermediate noisy states, conditioned on the current state $\mathbf{x}_n^0$. Following the formulation introduced by Ho et al. (2020), the reverse process is defined as a Markov chain:

$$p_\theta(\mathbf{x}_{n+1}^{0:T}) = p(\mathbf{x}_{n+1}^T) \prod_{t=1}^{T} p_\theta(\mathbf{x}_{n+1}^{t-1} \mid \mathbf{x}_{n+1}^t), \tag{4}$$

where each reverse transition is parameterised as a Gaussian distribution:

$$p_\theta(\mathbf{x}_{n+1}^{t-1} \mid \mathbf{x}_{n+1}^t) = \mathcal{N}\left(\mathbf{x}_{n+1}^{t-1}; \boldsymbol{\mu}_\theta(\mathbf{x}_{n+1}^t, \mathbf{x}_n^0, t), (\sigma^t)^2 \mathbf{I}\right). \tag{5}$$

The mean $\boldsymbol{\mu}_\theta(\mathbf{x}_{n+1}^t, \mathbf{x}_n^0, t)$ defines the centre of the reverse transition distribution $p_\theta(\mathbf{x}_{n+1}^{t-1} \mid \mathbf{x}_{n+1}^t)$, which aims to estimate the clean sample $\mathbf{x}_{n+1}^0$ by progressively denoising the noisy latent state $\mathbf{x}_{n+1}^t$ over multiple steps. This mean is not directly predicted by the neural network; rather, it is derived from the predicted noise component $\boldsymbol{\epsilon}_\theta(\mathbf{x}_{n+1}^t, \mathbf{x}_n^0, t)$, which approximates the Gaussian noise added to the clean sample during the forward process. The model is trained to estimate this noise using a





simple regression loss, and the predicted noise is then used during sampling to reconstruct the reverse mean via a closed-form expression derived from the posterior of the diffusion process:

$$\boldsymbol{\mu}_\theta(\mathbf{x}_{n+1}^t, \mathbf{x}_n^0, t) = \frac{1}{\sqrt{\alpha^t}} \left( \mathbf{x}_{n+1}^t - \frac{\beta^t}{\sqrt{1-\bar{\alpha}^t}} \boldsymbol{\epsilon}_\theta(\mathbf{x}_{n+1}^t, \mathbf{x}_n^0, t) \right), \tag{6}$$

where $\alpha^t$ and $\bar{\alpha}^t$ are scalar coefficients derived from a predefined forward noise schedule following (Ho et al., 2020), with $\alpha^t = 1 - \beta^t$ and $\bar{\alpha}^t = \prod_{s=1}^t \alpha^s$. The variance $(\sigma^t)^2$ determines the stochasticity in the reverse process and is typically set

to match the forward noise scale, i.e., $(\sigma^t)^2 = \beta^t$. In summary, $\boldsymbol{\epsilon}_\theta$ denotes the output of a neural network trained to predict the noise added during the forward process, while $\boldsymbol{\mu}_\theta$ is a deterministic function derived from $\boldsymbol{\epsilon}_\theta$ that defines the mean of the denoising Gaussian distribution used during inference. This parameterisation allows efficient training through a simple regression objective and enables high-quality sample generation during the reverse diffusion process.

### 3.1.3 Diffusion model training

To train a diffusion model capable of predicting the wildfire state at a future time point, a supervised learning strategy is employed. Consistent with the framework introduced by (Ho et al., 2020), the forward process is defined by a sequence of latent variables $\mathbf{x}_{n+1}^t$, where noise is gradually added to the clean wildfire state $\mathbf{x}_{n+1}^0$ across timesteps. At each pseudo-timestep $t$, the noisy version $\mathbf{x}_{n+1}^t$ is constructed as

$$\mathbf{x}_{n+1}^t = \sqrt{\bar{\alpha}^t}\, \mathbf{x}_{n+1}^0 + \sqrt{1-\bar{\alpha}^t}\, \boldsymbol{\epsilon}, \tag{7}$$

where $\boldsymbol{\epsilon} \sim \mathcal{N}(0, \mathbf{I})$ and $\bar{\alpha}^t$ is the cumulative product of the noise retention factors up to step $t$. In this context, $\mathbf{x}_{n+1}^t$ is the noisy representation of the state at pseudo-timestep $t$.

During training, the noise predictor $\boldsymbol{\epsilon}_\theta(\mathbf{x}_{n+1}^t, \mathbf{x}_n^0, t)$ is optimised to estimate the noise $\boldsymbol{\epsilon}$ that was introduced to the data frame at each pseudo-timestep $t$. The parameter set $\theta$ represents the trainable weights of the neural network, which serves as the noise predictor within the diffusion model. The objective of training is to minimise the discrepancy between the true noise $\boldsymbol{\epsilon}$, sampled from a standard Gaussian distribution $\mathcal{N}(0, \mathbf{I})$, and the predicted noise $\boldsymbol{\epsilon}_\theta(\mathbf{x}_{n+1}^t, \mathbf{x}_n^0, t)$, which is the model's

estimate of the noise present at pseudo-timestep $t$. By minimising this discrepancy, the model learns to progressively refine noisy representations $\mathbf{x}_{n+1}^t$ and reconstruct the original wildfire state $\mathbf{x}_{n+1}^0$.

The training objective can be reformulated into a simplified form that directly optimises the noise predictor $\boldsymbol{\epsilon}_\theta$. As introduced by Ho et al. (2020), this simplified objective measures the discrepancy between the true noise $\boldsymbol{\epsilon}$ and its predicted counterpart

$\boldsymbol{\epsilon}_\theta$. Incorporating the conditioning input $\mathbf{x}_n^0$, the simplified loss function is expressed as

$$\mathcal{L}_{\text{simple}}(\theta) = \mathbb{E}_{t, \boldsymbol{\epsilon}, \mathbf{x}_n^0} \left[ \left\| \boldsymbol{\epsilon} - \boldsymbol{\epsilon}_\theta(\mathbf{x}_{n+1}^t, \mathbf{x}_n^0, t) \right\|^2 \right]. \tag{8}$$

Following the noise estimation strategy, the training objective is to teach the model to recover the Gaussian noise that was used to perturb the clean wildfire state during the forward process. This is achieved by training a neural network to approximate the mapping from a noisy sample $\mathbf{x}_{n+1}^t$, timestep $t$, and conditioning input $\mathbf{x}_n^0$, to the original noise $\boldsymbol{\epsilon}$. In each

training iteration, a clean wildfire state $\mathbf{x}_{n+1}^0$ is first sampled from the data distribution $q(\mathbf{x}_{n+1}^0)$; in practice, this corresponds



to randomly selecting a wildfire frame from the training dataset generated by the CA model. A timestep $t$ is then uniformly sampled, and Gaussian noise $\epsilon \sim \mathcal{N}(\mathbf{0}, \mathbf{I})$ is added to the clean state according to the forward process to construct the noisy input $\mathbf{x}_{n+1}^t$. The model receives $\mathbf{x}_{n+1}^t$, the conditioning wildfire state $\mathbf{x}_n^0$, and the timestep $t$, and is trained to minimise the discrepancy between the true noise $\epsilon$ and its prediction $\epsilon_\theta(\mathbf{x}_{n+1}^t, \mathbf{x}_n^0, t)$. This formulation transforms the training task into a

denoising problem and enables the model to learn the reverse diffusion process in a supervised manner. The complete training procedure is summarised in Algorithm 1. Further theoretical background and justification for this approach are provided in Ho et al. (2020).

---

**Algorithm 1** Training of the noise predictor $\epsilon_\theta(\mathbf{x}_{n+1}^t, \mathbf{x}_n^0, t)$

---

1: **Input:** total number of pseudo-timesteps $T$

2: **repeat**

3:      $\mathbf{x}_{n+1}^0 \sim q(\mathbf{x}_{n+1}^0)$

4:      $t \sim \text{Uniform}(\{1, \ldots, T\})$

5:      $\epsilon \sim \mathcal{N}(\mathbf{0}, \mathbf{I})$

6:      Take gradient descent step on

$$\left\| \epsilon - \epsilon_\theta\left( \sqrt{\bar{\alpha}^t}\, \mathbf{x}_{n+1}^0 + \sqrt{1 - \bar{\alpha}^t}\, \epsilon,\, t,\, \mathbf{x}_n^0 \right) \right\|^2$$

7: **until** converged

---

Theoretically, the trained model should be able to generate a plausible prediction of the $(n+1)$-th frame based on the $n$-th frame.

### 3.1.4 Model inference

In this study, the DDIM algorithm (Song et al., 2022) is specifically adapted for predicting the future state of a wildfire. Given the current state $\mathbf{x}_n^0$ the model utilises the learned noise predictor $\epsilon_\theta(\mathbf{x}_{n+1}^{\tau_i}, \mathbf{x}_n^0, \tau_i)$ to estimate and remove noise, thereby generating a prediction for the future fire state $\mathbf{x}_{n+1}^0$. The modified DDIM reverse process for this application is represented as:

$$\mathbf{x}_{n+1}^{\tau_{i-1}} = \sqrt{\alpha^{\tau_{i-1}}} \left( \frac{\mathbf{x}_{n+1}^{\tau_i} - \sqrt{1 - \alpha^{\tau_i}}\, \epsilon_\theta}{\sqrt{\alpha^{\tau_i}}} \right) + \sqrt{1 - \alpha^{\tau_{i-1}} - \sigma^{\tau_i\, 2}} \cdot \epsilon_\theta + \sigma^{\tau_i} \mathbf{z}^{\tau_i} \tag{9}$$

where $\tau$ is an ascending sub-sequence sampled from $[1, \ldots, T]$ and $\mathbf{z}^{\tau_i}$ is standard Gaussian noise independent of $\mathbf{x}_{n+1}^{\tau_i}$. The term $\sigma^{\tau_i}$ represents the level of stochasticity introduced at each denoising step, controlling the magnitude of the noise $\mathbf{z}^{\tau_i} \sim \mathcal{N}(\mathbf{0}, \mathbf{I})$ added during the reverse process. Each inference process represents the model's attempt to predict the next frame $\mathbf{x}_{n+1}^0$ using the current frame $\mathbf{x}_n^0$ as a conditioning input. The detailed algorithm for the DDIM process is outlined in

Algorithm 2.




---

**Algorithm 2** DDIM sampling, given the noise predictor $\epsilon_\theta(\mathbf{x}_{n+1}^{\tau_i}, \mathbf{x}_n^0, \tau_i)$

---

1: **Input:** fire frame $\mathbf{x}_n^0$, total number of pseudo-timesteps $T$, total number of sampling steps $S$, noise weight $\eta$

2: $\mathbf{x}_{n+1}^T \sim \mathcal{N}(\mathbf{0}, \mathbf{I})$

3: $\tau \leftarrow \texttt{timesteps}(T, S)$

4: **for** $i = \tau_S, \ldots, \tau_1$ **do**

5: $\quad \sigma^{\tau_i} \leftarrow \eta \sqrt{\frac{1 - \alpha^{\tau_{i-1}}}{1 - \alpha^{\tau_i}}} \sqrt{1 - \frac{\alpha^{\tau_i}}{\alpha^{\tau_{i-1}}}}$

6: $\quad \mathbf{z} \sim \mathcal{N}(\mathbf{0}, \mathbf{I})$ if $t > 1$, else $\mathbf{z} = 0$

7: $\quad \epsilon_\theta \leftarrow \epsilon_\theta(\mathbf{x}_{n+1}^{\tau_i}, \mathbf{x}_n^0, \tau_i)$

8: $\quad \mathbf{x}_{n+1}^{\tau_{i-1}} \leftarrow \sqrt{\alpha^{\tau_{i-1}}} \left( \frac{\mathbf{x}_{n+1}^{\tau_i} - \sqrt{1 - \alpha^{\tau_i}} \epsilon_\theta}{\sqrt{\alpha^{\tau_i}}} \right) + \sqrt{1 - \alpha^{\tau_{i-1}} - \sigma^{\tau_i 2}} \cdot \epsilon_\theta + \sigma^{\tau_i} \mathbf{z}$

9: **end for**

10: **return** $\mathbf{x}_{n+1}^0$

---

The Denoising Diffusion Implicit Models (DDIM) framework accelerates inference in diffusion models through a non-Markovian reverse process that allows for larger sampling steps and improved efficiency (Song et al., 2022). Unlike standard DDPMs, which inject random noise at every timestep, DDIM enables sample generation through a noise-controlled mapping based solely on the initial latent variable $\mathbf{x}^T$. The hyperparameter $\eta$ modulates the level of stochasticity in the reverse process: setting $\eta = 1$ recovers the full randomness of the DDPM sampling procedure (Ho et al., 2020), while setting $\eta = 0$ eliminates per-step noise injection. In this study, we adopt DDIM with $\eta = 0$ by default to reduce sampling variance while still supporting diverse scenario generation through resampling of $\mathbf{x}^T$.

## 3.2 Model architecture

The noise predictor $\epsilon_\theta$ in our diffusion model employs a U-Net architecture (Ronneberger et al., 2015; Ho et al., 2020; Maji et al., 2022), which integrates both residual blocks and attention blocks to enhance performance in generating accurate predictions. The entire network structure is depicted in Figure 5, which illustrates two fundamental building blocks: the residual block and the attention block.





**Figure 5.** (a) Attention Res-UNet architecture; (b) residual block architecture; (c) attention block architecture

The residual convolutional block (He et al., 2015) constitutes a foundational element of the network architecture, engineered to enhance the efficacy and reliability of the training process by facilitating the transmission of gradients throughout the network.This block integrates several pivotal components, including convolutional layers, group normalization, the SiLU activation function, and skip connections. The incorporation of these elements ensures that the residual block effectively captures and propagates critical information while mitigating issues such as vanishing gradients. The skip connections, in particular, play a crucial role in preserving information across layers, thereby enhancing the robustness of the learning process. The attention block (Vaswani et al., 2017), on the other hand, is important for enabling the model to focus on specific aspects of the




input data, thereby enhancing its ability to represent intricate features. For a detailed description of the full architecture and the arrangement of residual and attention components within the network, please refer to Appendix D.

### 3.3 Ensemble sampling

In this study, an ensemble sampling method is employed to leverage the randomness introduced during the inference process of the diffusion model. This approach allows the model to generate a range of potential outputs for a given input, capturing the
variability and uncertainty inherent in wildfire spread predictions.

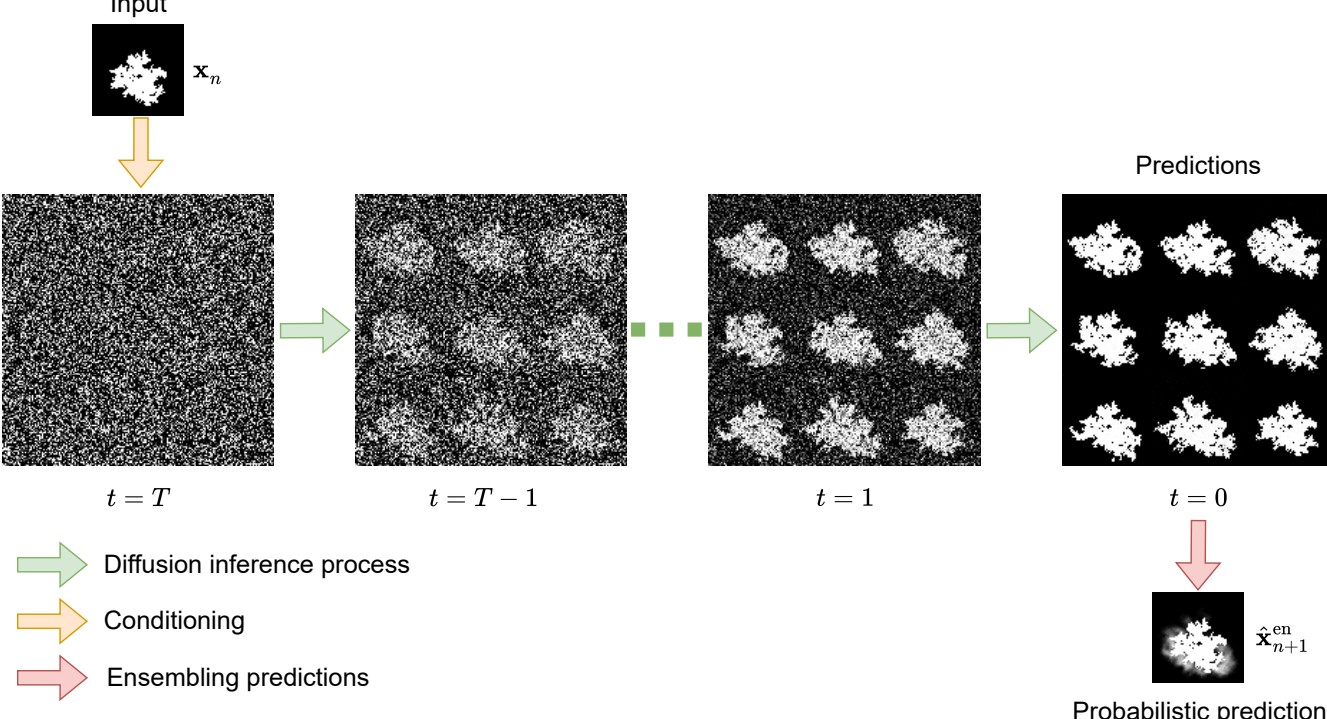

**Figure 6.** Generate probabilistic prediction undergoes multiple diffusion inference processes.





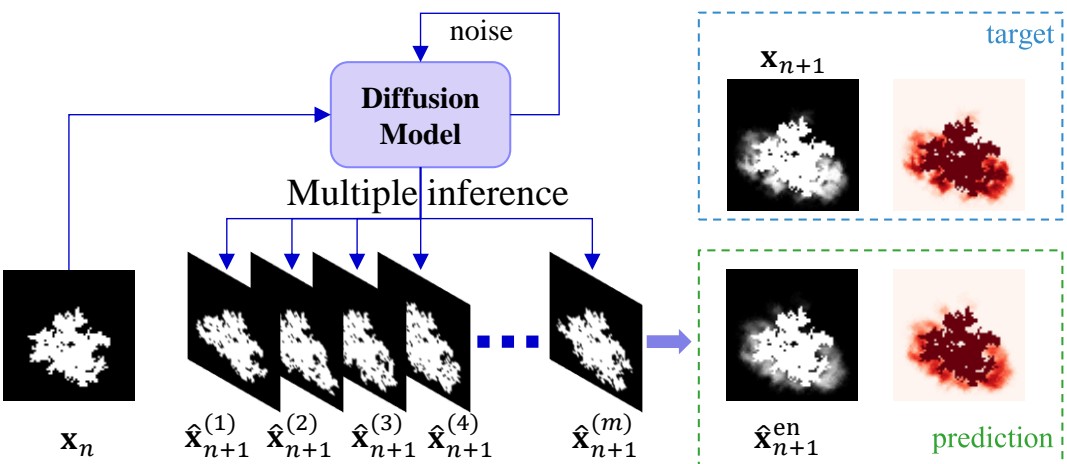

**Figure 7.** Model evaluation.

The ensemble sampling method involves performing multiple inference passes for the same input fire state frame, $\mathbf{x}_n^0$, to account for the inherent uncertainty in wildfire spread. Under the DDIM framework (Song et al., 2022), although the reverse process is noise-free at each timestep when $\eta = 0$, randomness is retained through the sampling of the initial latent variable $\mathbf{x}^T$. As a result, each inference run with a different noise seed can produce a distinct prediction of the next frame, denoted as $\widehat{\mathbf{x}}_{n+1}^0$. As shown in Figure 6, to generate a probabilistic forecast rather than a single prediction, the model is executed $M$ times for the same input frame, yielding a set of predictions $\{\widehat{\mathbf{x}}_{n+1}^{0,(1)}, \widehat{\mathbf{x}}_{n+1}^{0,(2)}, \ldots, \widehat{\mathbf{x}}_{n+1}^{0,(M)}\}$. By aggregating and averaging these outputs, an ensemble prediction is obtained, offering a more robust estimate of the underlying fire spread dynamics while capturing the variability inherent in the wildfire propagation process.

These predictions differ due to the stochasticity embedded in the diffusion model, ensuring that the ensemble captures a range of possible outcomes. As illustrated in Figure 7, the ensemble prediction is computed by aggregating the results of these multiple inference passes. Specifically, the predictions are averaged to produce the ensemble output,

$$\widehat{\mathbf{x}}_{n+1}^{\mathrm{en}} = \frac{1}{M} \sum_{m=1}^{M} \widehat{\mathbf{x}}_{n+1}^{0,(m)}. \tag{10}$$

This aggregation step leverages the diversity of individual predictions to produce a robust and representative estimate of the next frame. The final output, $\widehat{\mathbf{x}}_{n+1}^{\mathrm{en}}$ is a stochastic prediction designed to better capture the variability and uncertainty inherent in the wildfire spread process. In contrast to deterministic models, which yield a single, fixed outcome, the ensemble sampling method incorporates the variability present in multiple inference passes, resulting in a probabilistic forecast that more effectively represents the spectrum of potential outcomes.



## 4 Experiments

### 4.1 Experiments design

To evaluate the performance of the diffusion model in capturing the stochasticity of wildfire spread events compared to deterministic models, a series of comparative experiments were conducted. The objective was to demonstrate the advantages of the diffusion model in probabilistic wildfire modelling, particularly in the context of the complex and unpredictable nature associated with wildfire propagation. To ensure a fair comparison, both the diffusion model and a deterministic benchmark model utilised the same neural network architecture—the attention Res-UNet described in Section 2.3. The deterministic model was

trained within a conventional supervised learning setting, explicitly predicting the subsequent wildfire state by minimising the mean squared error (MSE) loss between the predicted and ground truth burned areas given by the fire simulation. The diffusion model followed a conditional diffusion framework, which learns to generate probabilistic predictions by gradually refining noisy inputs through a denoising process, optimising an MSE loss between the predicted and target noise distributions at each diffusion step. This setup allows for a direct comparison of the models' ability to predict wildfire spread dynamics. Since both

models share an identical architecture but differ in their training strategies—one employing deterministic regression and the other conditional diffusion—any observed performance differences can be attributed to the diffusion model's ability to capture the stochastic nature of wildfire spread more effectively.

The data utilised for these experiments were generated using a CA simulator, informed by observational records collected from the Chimney fire and Ferguson fire incidents. For each of these fire events, separate training and ensemble testing datasets were created to ensure the robustness and independence of the model evaluation process. Model training was conducted exclu-

sively on the training dataset, while performance evaluation was carried out independently using the ensemble testing dataset.

The characteristics of these datasets are summarised in Table 2. Here, the term *dataset size* refers to the total number of wildfire spread scenarios (individual data samples) contained within each respective dataset.

|  | Training dataset | Ensemble testing dataset |
| --- | --- | --- |
| Resolution | $64 \times 64$ | $64 \times 64$ |
| Dataset size | 900 | 50 |
| Ensemble size | – | 50 |

**Table 2.** Summary of training and ensemble testing datasets

To assess the performance of the models, a range of evaluation metrics were employed. Mean squared error (MSE) was

used to measure the average squared difference between the predicted wildfire spread, denoted as $\widehat{x}_{n+1}$, and the actual wildfire spread, denoted as $x_{n+1}$, where $x_{n+1}$ represents the burned area generated from the CA simulation which are considered as ground truth in these experiments. Lower MSE values indicate better predictive accuracy. The Peak signal-to-noise ratio (PSNR) was adopted to quantify the fidelity of predicted wildfire spread maps relative to ground truth observations, where higher values reflect better preservation of critical spatial details at the pixel level. The Structural similarity index measure



(SSIM) evaluated the perceptual coherence between predictions and reality, emphasizing how well the model preserved natural patterns in luminance, contrast, and spatial structure. These standard image similarity metrics were complemented by a custom-defined metric, the Hit rate (HR), which measures the proportion of correctly predicted burned regions within a predefined threshold. HR, defined in Equation (B1), considers only regions where fire spread has occurred in the ground truth data, reflecting the model's ability to accurately capture areas affected by wildfire. To further assess the realism of the generated

predictions, the Fréchet inception distance (FID) was employed to compare the feature distributions between the predicted and ground truth wildfire spread maps. It should be noted, however, that the FID metric relies on a feature extractor pre-trained on natural images from the ImageNet dataset. As such, it quantifies photorealism rather than geophysical realism. While FID does not directly measure the physical accuracy of wildfire spread, it serves as a useful proxy for evaluating the perceptual quality and statistical similarity of the generated outputs in a high-dimensional feature space. Kullback–Leibler divergence (KL) was

computed to quantify the difference between the predicted and actual probability distributions, with lower values indicating a closer match between the two distributions. These metrics collectively provide a comprehensive framework for evaluating the models' abilities to simulate and predict wildfire spread, capturing both their accuracy in terms of pixel-level predictions and their ability to represent the underlying uncertainty in wildfire dynamics. A full description of these evaluation metrics can be found in Appendix B.





## 4.2    Results and analysis

| | Deterministic | | | | | Diffusion | | | | |
|---|---|---|---|---|---|---|---|---|---|---|
| Size / Metric | 50 | 100 | 200 | 500 | 900 | 50 | 100 | 200 | 500 | 900 |
| MSE ↓ | 0.0199 | 0.0191 | 0.0160 | 0.0149 | 0.0143 | 0.0105 | 0.0077 | 0.0065 | 0.0056 | 0.0053 |
| PSNR ↑ | 17.149 | 17.260 | 18.171 | 18.460 | 18.543 | 17.820 | 21.419 | 22.080 | 22.873 | 23.127 |
| SSIM ↑ | 0.8348 | 0.8383 | 0.8444 | 0.8449 | 0.8467 | 0.5154 | 0.8365 | 0.8692 | 0.8923 | 0.8968 |
| HR ($\epsilon = 0.2$) ↓ | 0.7186 | 0.7231 | 0.7327 | 0.7350 | 0.7376 | 0.7360 | 0.8313 | 0.8495 | 0.8692 | 0.8754 |
| FID ↓ | 181.35 | 182.92 | 182.21 | 182.19 | 179.96 | 112.99 | 90.021 | 72.730 | 42.260 | 38.540 |
| KL ↓ | - | - | - | - | - | 341.45 | 206.96 | 201.20 | 173.80 | 169.80 |

(a) Performance on Chimney fire dataset

| | Deterministic | | | | | Diffusion | | | | |
|---|---|---|---|---|---|---|---|---|---|---|
| Size / Metric | 50 | 100 | 200 | 500 | 900 | 50 | 100 | 200 | 500 | 900 |
| MSE ↓ | 0.0323 | 0.0312 | 0.0298 | 0.0295 | 0.0292 | 0.0112 | 0.0094 | 0.0087 | 0.0085 | 0.0081 |
| PSNR ↑ | 14.906 | 15.054 | 15.253 | 15.309 | 15.343 | 19.523 | 20.271 | 20.599 | 20.710 | 20.929 |
| SSIM ↑ | 0.7766 | 0.8214 | 0.8146 | 0.8032 | 0.8205 | 0.6942 | 0.8474 | 0.8368 | 0.8447 | 0.8601 |
| HR ($\epsilon = 0.2$) ↓ | 0.7258 | 0.7163 | 0.7150 | 0.7301 | 0.7216 | 0.7882 | 0.8052 | 0.8095 | 0.8150 | 0.8184 |
| FID ↓ | 161.04 | 151.72 | 168.07 | 168.33 | 151.76 | 122.47 | 86.402 | 94.310 | 64.106 | 59.462 |
| KL ↓ | - | - | - | - | - | 289.05 | 308.85 | 273.72 | 235.01 | 252.01 |

(b) Performance on Ferguson fire dataset

**Table 3.** Comparative performance of deterministic and diffusion models across various training set sizes on multiple metrics. Size represents the size of the training dataset, with values given in the number of 50, 100, 200, 500, and 900 samples; Deterministic refers to the performance of the benchmark model; Metrics including MSE, PSNR, SSIM, FID, KL and HR.

The results of the comparative experiments between the deterministic benchmark model and the diffusion model for the Chimney fire and Ferguson fire events are presented independently in Table 3. Both models were evaluated separately for each wildfire scenario using identical metrics across various training dataset sizes. In each independent experiment, the diffusion model employed the DDIM sampling method, configured with 600 total number of pseudo-timesteps ($T$) and a total number of sampling steps ($S$) of 50, which were uniformly skipped based on a linear scheduling strategy, evenly distributing selected sampling steps across the entire sampling trajectory. This experimental setup ensured an optimal balance between sampling





efficiency and predictive accuracy, enabling the reliable generation of high-quality probabilistic predictions within practical computational constraints.

### 4.2.1 Overall performance

The diffusion model has been demonstrated to outperform the benchmark deterministic model across a range of metrics and training dataset sizes on both the Chimney and the Ferguson fire cases as shown in Table 1. Specifically, the diffusion model exhibits considerable improvements in metrics that evaluate the similarity between predicted and actual probability distributions, including PSNR, SSIM, FID, and KL. These metrics are particularly important for stochastic models, as they indicate the model's ability to generate predictions that closely resemble the reference simulations, even in the presence of randomness

and variability. In this experiment, this is evidenced by the ensemble predictions generated by the diffusion model being more closely aligned with the ground truth distribution compared to the deterministic predictions.

The systematic evaluation across training dataset sizes (50, 100, 200, 500, and 900 samples) reveals distinct scaling properties between architectures. For the Chimney fire dataset, the diffusion model achieves superior data efficiency, reducing mean squared error (MSE ↓) by $49.5\%$ ($0.0105 \rightarrow 0.0053$), compared to the deterministic model's reduction of $28.1\%$ ($0.0199 \rightarrow$

$0.0143$) when scaling from 50 to 900 samples. This performance gap is further amplified in distribution-sensitive metrics: the diffusion model's peak signal-to-noise ratio (PSNR ↑) improves by $5.307$ dB ($17.82 \rightarrow 23.127$), representing a $29.8\%$ relative gain versus the deterministic model's modest increase of $8.1\%$ ($17.149 \rightarrow 18.543$).

The structural similarity index (SSIM) exhibits parallel trends, with diffusion models achieving $74.1\%$ greater improvement ($0.5154 \rightarrow 0.8968$) compared to deterministic baselines ($0.8348 \rightarrow 0.8467$) at maximal training size. Distributional metrics

further confirm this advantage: the diffusion model's Fréchet Inception Distance (FID ↓) decreases significantly by $65.9\%$ ($112.99 \rightarrow 38.54$), whereas the deterministic model shows negligible improvement of $1.0\%$ ($181.35 \rightarrow 179.96$). Notably, the Kullback-Leibler divergence (KL ↓) was computed exclusively for the diffusion model, highlighting a consistent and substantial reduction of $50.3\%$ ($341.45 \rightarrow 169.80$) as the training size increased from 50 to 900 samples.

These observed patterns persist in the independent Ferguson fire dataset experiments, where the diffusion model demon-

strates similarly strong scaling behaviour. Each doubling of the training dataset size yields approximately $12\%$ greater MSE reductions compared to deterministic baselines, culminating in an absolute SSIM improvement of $27.7\%$ ($0.6942 \rightarrow 0.8601$), markedly surpassing the deterministic improvement of only $5.6\%$ ($0.7766 \rightarrow 0.8205$).

The hit rate (HR ↓) metric, calculated using a threshold of $\epsilon = 0.2$ (see Equation (B1) for details), confirms enhanced spatial accuracy: diffusion models achieve absolute improvements of $+13.9\%$ ($73.6\% \rightarrow 87.5\%$) and $+3.0\%$ ($72.2\% \rightarrow 81.8\%$) for

Chimney and Ferguson fires respectively at $N = 900$, clearly outperforming deterministic benchmarks which plateau around $73.8\%$ and $72.2\%$.

The diffusion-based stochastic model consistently outperforms the deterministic benchmark across all evaluation metrics, demonstrating superior data efficiency, better performance on distribution-sensitive metrics, and greater scalability with larger training datasets. This trend suggests that the conditional diffusion training strategy is more effective in leveraging larger

datasets to refine its predictions. The diffusion model's ability to learn from additional data enables it to capture new patterns





and relationships, enhancing its capacity to predict future wildfire states. By contrast, the deterministic model shows limited improvement as the dataset size increases, indicating a more constrained learning capacity under the same architectural framework. Overall, the diffusion model excels in capturing the stochastic nature of wildfire spread, offering more accurate and probabilistic predictions compared to the deterministic benchmark.





**4.2.2 Impact of diffusion sampling times**



**Figure 8.** Evaluation of number of diffusion sampling iterations on performance metrics across different fire datasets: (1)∼(4) correspond to the Chimney Fire (2016) dataset; (5)∼(8) correspond to the Ferguson Fire (2018) dataset. The subplots illustrate the impact of sampling times on various performance metrics, including MSE, PSNR, SSIM, FID, KL and HR. The training dataset for each fire scenario consists of 900 samples.





Figure 8 evaluates how the number of diffusion sampling iterations—defined as the number of inference iterations performed by the diffusion model to generate a single ensemble prediction—affects various performance metrics across different wildfire datasets. The results demonstrate a clear trend: increasing the number of diffusion sampling iterations generally improves prediction quality. This trend is consistent across both examined datasets: the Chimney fire dataset, as shown in subplots (1)

to (4), and the Ferguson fire dataset, represented by subplots (5) to (8). When the number of diffusion sampling iterations is set to 1, the generated predictions are essentially binary. As the number of diffusion sampling iterations increases from 1 to 50, several performance metrics consistently improve. Mean squared error (MSE) and Fréchet inception distance (FID) decrease, indicating that the generated predictions become more similar to the ground truth wildfire spread maps in both numerical accuracy and feature similarity. Peak signal-to-noise ratio (PSNR) and Structural similarity index measure (SSIM)

increase, suggesting that the generated wildfire states exhibit greater structural coherence and perceptual quality compared to the deterministic baseline. Additionally, the hit rate (HR) improves, reflecting better consistency in predicting burned and unburned regions.

The visualisations in Figure 9 compare the ensemble predictions and deterministic predictions across six scenarios, with the first three ((a)∼(c)) derived from the Chimney fire test dataset and the latter three ((d)∼(f)) from the Ferguson fire test dataset.

Each scenario is divided into three sections: the leftmost column shows the input ($\mathbf{x}_n$), target ($\mathbf{x}_{n+1}$), ensemble prediction ($\widehat{\mathbf{x}}_{n+1}^{\text{en}}$) and deterministic prediction ($\widehat{\mathbf{x}}_{n+1}^{\text{det}}$); the middle five columns display mismatch plots illustrating true positives (green), false positives (red), and false negatives (blue) across thresholds ranging from $0.2$ to $0.8$; and the rightmost column performance metrics (F1 score, F2 score, precision, and MCC) as functions of the threshold.

In this context, the threshold refers to the cutoff value applied to pixel intensity in the predicted fire probability map, where

pixel values greater than the threshold are classified as positive (predicted wildfire spread), and those below are classified as negative. This thresholding approach allows us to explore the model's performance under varying levels of sensitivity to fire prediction. A higher threshold means that only areas with a higher probability of fire spread are considered as predicted fire regions, while a lower threshold includes more areas, increasing the number of predicted positive fire regions. This helps in assessing how the model performs under different levels of confidence and sensitivity.

These thresholds have been applied to both predictions and targets from the diffusion and deterministic models to ensure a fair comparison. In wildfire spread forecasting, different thresholds are used to explore the model's sensitivity at various levels of confidence. At low thresholds, the model is more sensitive, capturing more regions as likely fire spread, which increases recall but also increases false positives. This is important for detecting all possible areas that may be at risk, even if it means some false predictions are included. As the threshold increases, the model becomes less sensitive, focusing on areas that are

more certainly predicted as fire spread, which may reduce false positives but also potentially miss smaller, less intense fires or fire spread in areas with more uncertainty. This trade-off is crucial when managing wildfire risk, as it determines the balance between detecting all potential fire spread (higher recall) and ensuring that predictions are precise (minimising false positives).

In the mismatch plots (as shown in the middle five columns of Figure 9), the effect of varying thresholds is clearly illustrated. Ensemble predictions remain relatively stable across thresholds, with consistent performance in terms of true positives, false

positives, and false negatives. This stability suggests that ensemble models are better at maintaining accuracy under different





levels of sensitivity, making them more reliable for forecasting wildfire spread under varying conditions. By contrast, deterministic predictions show higher sensitivity to threshold changes, particularly at lower thresholds, where they tend to over-predict fire spread, leading to more false positives. As the threshold increases, deterministic models become more selective, reducing false positives but also decreasing recall, missing important areas where the fire could potentially spread.

In the performance metric plots (the rightmost column of Figure 9), we observe that ensemble predictions maintain stable scores for all metrics, with a noticeable increase in F1 score and precision at higher thresholds (e.g., $> 0.5$). The ensemble method's ability to balance recall and precision allows it to achieve consistently better performance than deterministic models, especially when a higher threshold is needed to focus on more confident predictions. By contrast, deterministic predictions exhibit a significant drop in precision as the threshold increases, suggesting that they are less robust and more prone to false

negatives when the threshold is set higher. The F1 score, which balances both precision and recall, is particularly important in wildfire prediction. A higher F1 score at higher thresholds indicates that the ensemble model can more accurately predict fire spread without sacrificing recall for precision. The precision metric, in particular, shows the greatest difference between ensemble and deterministic predictions at higher thresholds, where ensemble predictions maintain high precision, making them more suitable for decision-making processes that require accurate identification of fire-prone areas. The Matthews correlation

coefficient (MCC), a more robust metric that accounts for both false positives and false negatives, demonstrates the ensemble model's superior performance at all threshold levels. The ensemble model consistently achieves higher MCC scores, indicating that it provides a more reliable and balanced forecast of wildfire spread. These results are consistent with those presented in Table 3, reinforcing the advantages of the ensemble model in capturing the uncertainty and variability in wildfire spread dynamics compared to the deterministic approach.







**Figure 9.** Visual comparison of ensemble predictions and deterministic predictions for wildfire spread, with (a)∼(c) derived from the Chimney fire (2016) test dataset and (d)∼(f) from the Ferguson fire (2018) test dataset.





## 5 Conclusions

This study proposes a stochastic framework for wildfire spread prediction using conditional denoising diffusion models, aiming to build a generative surrogate that captures uncertainty and spatial variability for probabilistic forecasting. Unlike conventional deterministic models that generate a single forecast, our diffusion-based emulator produces an ensemble of plausible future states through repeated sampling from a generative process conditioned on the observed fire state. This ensemble-based formulation yields probabilistic forecasts that better reflect the inherent stochasticity of wildfire spread and are thus more informative for risk-aware decision-making. Trained on synthetic wildfire data generated by a probabilistic cellular automata simulator incorporating real environmental factors, the proposed model outperforms a deterministic baseline with identical architecture across multiple performance metrics. In both the Chimney and Ferguson fire datasets, the diffusion model demonstrates significantly lower prediction error and stronger structural fidelity, while also producing distributions that more closely match the reference simulations. These improvements are particularly evident in distribution-sensitive metrics such as the Fréchet Inception Distance, highlighting the model's ability to recover complex spatial features. Moreover, the ensemble predictions show enhanced stability and robustness under threshold variation, providing a consistent and reliable basis for downstream risk analysis. Together, these findings underscore the suitability of diffusion-based emulators for probabilistic wildfire modelling and point to their broader potential in geophysical forecasting applications.

Future work will aim to enhance the model's generalisability by incorporating climate forcings and geophysical features, including meteorological variables, terrain properties, and vegetation structure, as additional conditioning inputs. Achieving this requires exposure to a wider diversity of fire regimes and environmental settings, which in turn motivates the development of more extensive multi-region training datasets. Integrating such contextual information is expected to improve the model's adaptability across heterogeneous landscapes and increase its practical applicability in operational wildfire forecasting and long-term planning.



## Appendix A: Notation table

| **Main Notations** | |
|---|---|
| **Notation** | **Description** |
| $\mathbf{x}_n$ | The current state of the wildfire, represented as a fire state frame, where each element indicates the fire's state at a specific location. |
| $\mathbf{x}_{n+1}$ | The next fire state frame, representing the wildfire spread at the next time step. |
| $\widehat{\mathbf{x}}_{n+1}^{\text{en}}$ | The ensemble prediction of the next fire state, obtained by aggregating multiple predictions generated by the diffusion model. |
| $\widehat{\mathbf{x}}_{n+1}^{\text{det}}$ | The deterministic prediction of the next fire state, generated by the deterministic model. |
| $T$ | The total number of pseudo-timesteps used in the diffusion model. |
| $\mathbf{x}_{n+1}^{t,(m)}$ | The $m$-th sample generated at pseudo-timestep $t$ during the diffusion process, representing a predicted fire state at a specific point in time. |
| $\theta$ | The trainable parameters of the noise predictor within the diffusion model. |
| $\boldsymbol{\epsilon}$ | Standard Gaussian noise, $\boldsymbol{\epsilon} \sim \mathcal{N}(\mathbf{0}, \mathbf{I})$ |
| $\boldsymbol{\epsilon}_\theta(\mathbf{x}_{n+1}^t, \mathbf{x}_n^0, t)$ | Model-predicted noise given $\mathbf{x}^t$ and timestep $t$ |
| $\boldsymbol{\mu}_\theta(\mathbf{x}_{n+1}^t, \mathbf{x}_n^0, t)$ | Mean of the reverse process distribution at timestep $t$ |
| $q(\cdot)$ | The approximate posterior distribution used in the forward (diffusion) process. |
| $p_\theta(\cdot)$ | The predicted posterior distribution in the backward (denoising) process, parameterized by $\theta$. |
| $\beta^t$ | The variance schedule used to define the noise added at each pseudo-timestep in the diffusion process. |
| $\alpha^t$ | The scaling factor in the noise schedule, used in the forward and backward process, is defined as $\alpha^t = 1 - \beta^t$. |
| $\sigma^t$ | The noise level at pseudo-timestep $t$, typically defined by $\sigma^t = \sqrt{\beta^t}$, used in the backward process. |

Table A1: Main notations





## Appendix B: Evaluation metrics

- **Mean squared error (MSE)**: A standard loss function that measures the average squared difference between the predicted and actual values. Lower MSE values indicate better performance.

- **Peak signal-to-noise ratio (PSNR)**: A metric that quantifies the ratio between the maximum possible power of a signal and the power of corrupting noise. A higher PSNR value indicates a superior reconstruction quality (Ho et al., 2020).

- **Structural similarity index measure (SSIM)**: A perceptual metric that assesses the similarity between two images based on luminance, contrast, and structure. The SSIM value ranges from -1 to 1, with higher values indicating a greater degree of similarity (Ho et al., 2020).

- **Hit rate (HR)**: The HR metric is defined as the ratio of correctly predicted positive instances within a specified threshold. Let $\widehat{\mathbf{x}}_{n+1}$ represent the predicted wildfire spread and $\mathbf{x}_{n+1}$ the ground truth wildfire state. The metric only considers locations where the target $\mathbf{x}_{n+1}$ is greater than zero. The HR is defined in Equation (B1):

$$\mathrm{HR} = \frac{\sum_{\mathrm{valid}\ i} \mathbf{1}(|\widehat{\mathbf{x}}_{n+1,i} - \mathbf{x}_{n+1,i}| < \epsilon)}{N_{\mathrm{valid}}} \tag{B1}$$

where $\epsilon$ is the threshold that defines the acceptable difference between $\widehat{\mathbf{x}}_{n+1}$ and $\mathbf{x}_{n+1}$; $\mathbf{1}(\cdot)$ is the indicator function, which equals 1 when the condition inside it is true and 0 otherwise; and $N_{\mathrm{valid}}$ is the number of valid target elements where $\mathbf{x}_{n+1} > 0$.

- **Fréchet inception distance (FID)**: A metric that measures the similarity between two datasets of images, evaluating the quality of the generated images by comparing their feature distributions, and has been widely used to assess the image quality of generative models (Heusel et al., 2018).

- **Kullback–Leibler divergence (KL)**: A measure of the discrepancy between two probability distributions, whereby the divergence is quantified. A lower value of the Kullback–Leibler divergence indicates that the predicted distribution is closer to the ground truth distribution (Goldberger et al., 2003).



## Appendix C: Acronyms

**CA** Cellular automata

**MODIS** Moderate Resolution Imaging Spectroradiometer

**VIIRS** Visible Infrared Imaging Radiometer Suite

**DDPM** Denoising Diffusion Probabilistic Models

**DDIM** Denoising Diffusion Implicit Models

**VLB** Variational lower bound

**MSE** Mean squared error

**PSNR** Peak signal-to-noise ratios

**SSIM** Structural similarity index measure

**HR** Hit rate

**FID** Fréchet inception distance

**KL** Kullback–Leibler divergence

**MCC** Matthews correlation coefficient

**ML** Machine learning

**DL** Deep learning

**ESMs** Earth System Models



**Appendix D: Attention Res-UNet architecture**

| Stage | Layer Type | Output Shape | Kernel | In_Channels | Out_Channels | Param # |
|---|---|---|---|---|---|---|
| Input | Time Embedding | (1, 512) | 2×Linear + SiLU | 128 → 512 | 512 → 512 | 328,704 |
| | Conv2d (input conv) | (1, 128, 64, 64) | 3×3 | 2 | 128 | 2,432 |
| Down sample | ResidualBlock | (1, 128, 64, 64) | 3×3 | 128 | 128 | 361,344 |
| | ResidualBlock | (1, 128, 64, 64) | 3×3 | 128 | 128 | 361,344 |
| | Downsample | (1, 128, 32, 32) | 3×3 (stride=2) | 128 | 128 | 147,584 |
| | ResidualBlock | (1, 256, 32, 32) | 3×3 | 128 | 256 | 1,050,368 |
| | ResidualBlock | (1, 256, 32, 32) | 3×3 | 256 | 256 | 1,312,512 |
| | AttentionBlock | (1, 256, 32, 32) | 1×1(QKV)+1×1(Proj) | 256 | 256 | 262,912 |
| | Downsample | (1, 256, 16, 16) | 3×3 (stride=2) | 256 | 256 | 590,080 |
| | ResidualBlock | (1, 256, 16, 16) | 3×3 | 256 | 256 | 1,312,512 |
| | ResidualBlock | (1, 256, 16, 16) | 3×3 | 256 | 256 | 1,312,512 |
| | Downsample | (1, 256, 8, 8) | 3×3 (stride=2) | 256 | 256 | 590,080 |
| | ResidualBlock | (1, 256, 8, 8) | 3×3 | 256 | 256 | 1,312,512 |
| | ResidualBlock | (1, 256, 8, 8) | 3×3 | 256 | 256 | 1,312,512 |
| Bottleneck | ResidualBlock | (1, 256, 8, 8) | 3×3 | 256 | 256 | 1,312,512 |
| | AttentionBlock | (1, 256, 8, 8) | 1×1(QKV)+1×1(Proj) | 256 | 256 | 262,912 |
| | ResidualBlock | (1, 256, 8, 8) | 3×3 | 256 | 256 | 1,312,512 |
| Up sample | ResidualBlock (skip) | (1, 256, 8, 8) | 3×3 | 256+256 | 256 | 2,034,176 |
| | ResidualBlock | (1, 256, 8, 8) | 3×3 | 256 | 256 | 2,034,176 |
| | Upsample | (1, 256, 16, 16) | nearest + 3×3 | 256 | 256 | 590,080 |
| | ResidualBlock (skip) | (1, 256, 16, 16) | 3×3 | 256+256 | 256 | 2,034,176 |
| | ResidualBlock | (1, 256, 16, 16) | 3×3 | 256 | 256 | 2,034,176 |
| | Upsample | (1, 256, 32, 32) | nearest + 3×3 | 256 | 256 | 590,080 |
| | ResidualBlock (skip) | (1, 256, 32, 32) | 3×3 | 256+256 | 256 | 2,034,176 |
| | ResidualBlock | (1, 256, 32, 32) | 3×3 | 256 | 256 | 2,034,176 |
| | AttentionBlock | (1, 256, 32, 32) | 1×1(QKV)+1×1(Proj) | 256 | 256 | 262,912 |
| | Upsample | (1, 256, 64, 64) | nearest + 3×3 | 256 | 256 | 590,080 |
| | ResidualBlock (skip) | (1, 128, 64, 64) | 3×3 | 256+128 | 128 | 706,048 |
| | ResidualBlock | (1, 128, 64, 64) | 3×3 | 128 | 128 | 541,952 |
| Output | Conv2d + GN + SiLU | (1, 1, 64, 64) | 3×3 | 128 | 1 | 1,153 |
| | | | | | Total Param # | 84,049,793 |

**Table D1.** UNet Architecture



| Sub-Module | Architecture Details | Notes |
|---|---|---|
| **time_emb** | `SiLU`<br>`Linear`(*time_channels* → *out_channels*) | Applies activation to time embedding, |
| then projects to match out_channels. | | |
| **conv1** | `GroupNorm`(*norm_groups*, *in_channels*)<br>`SiLU`<br>`Conv2d`(*in_channels* → *out_channels*, 3×3, padding=1) | First residual path convolution. |
| **conv2** | `GroupNorm`(*norm_groups*, *out_channels*)<br>`SiLU`<br>`Dropout`(p=*dropout*)<br>`Conv2d`(*out_channels* → *out_channels*, 3×3, padding=1) | Second residual path convolution. |
| **shortcut** | `Identity` *(if in_channels = out_channels)*<br>`Conv2d`(*in_channels* → *out_channels*, 1×1) *(otherwise)* | Ensures shape match for the skip connection. |

**Table D2.** Structure of the ResidualBlock.

| Sub-Module | Architecture Details | Notes |
|---|---|---|
| **norm** | `GroupNorm`(*norm_groups*, *channels*) | Normalizes the input before QKV projections. |
| **qkv** | `Conv2d`(*channels* → 3×*channels*, 1×1, bias=False) | Generates query, key, and value embeddings. |
| **multi-head attn** | Splits Q, K, V into heads<br>Scaled dot-product, softmax<br>Combines attended vectors | Implements multi-head self-attention |
| on spatial positions. | | |
| **proj** | `Conv2d`(*channels* → *channels*, 1×1) | Final linear projection, |
| added back to the input (residual). | | |

**Table D3.** Structure of the AttentionBlock.

*Code and data availability.* The code and data underpinning this study are publicly available in a GitHub repository and have been permanently archived on Zenodo (Yu, 2025). These resources are accessible under the terms of the MIT licence, which permits free use, modification and redistribution. The actual values of the geological features, such as landscape slope, vegetation density, and vegetation cover, are obtained from the Interagency Fuel Treatment Decision Support System (IFTDSS) (Int, Thu, 02/01/2024 - 10:55) for corresponding ecoregions.

*Acknowledgements.* Wenbo Yu and Sibo Cheng acknowledges the support of the French Agence Nationale de la Recherche (ANR) under reference ANR-22-CPJ2-0143-01. CEREA is a member of Institut Pierre-Simon Laplace (IPSL).





*Author contributions.*

WY and AG performed the formal analysis of the data; WY, AG and SB performed the model simulations; WY and SB
prepared the manuscript with contributions from all coauthors; RA, MB and SB provided the financial support for the project
to led to this publication; RA, MB and SB coordinated research activities; TSF provided technicnal support; all co-authors
reviewed and edited the manuscript.

*Competing interests.*

The contact author has declared that none of the authors has any competing interests.



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
