# Peer review of "A Probabilistic Approach to Wildfire Spread Prediction Using a Denoising Diffusion Surrogate Model"

_EGUsphere, 2025_

## Author Comment (AC1)

**Reply to referee 1**

We thank the Reviewer for the comments and suggestions on our manuscript.

We have responded to most of the Reviewer's comments in our online reply dated 9 August 2025 (at the end of this letter, for reference). Following the Reviewer's suggestion, we conducted comprehensive numerical experiments using three different neural network architectures, namely a UNet with attention mechanism, a standard UNet, and a convolutional autoencoder with ResBlocks. Each network was trained both deterministically and using diffusion-based training. The numerical results are provided in Appendix D (p. 32) of the revised manuscript and are summarised below.

[Figure]

Figure 1: Ablation study comparing model performance across different training dataset sizes and architectures.

Our ablation study evaluates three distinct neural network architectures across varying training dataset sizes {50, 100, 200, 500, and 900 samples}, comparing stochastic diffusion models against deterministic approaches, as shown in Figure 1. The architectures under scrutiny include: (1) a UNet architecture with attention mechanisms (unet_atten) as employed in our

main study; (2) a UNet basic architecture (unet_basic), representing a simplified variant without attention blocks; and (3) a residual AutoEncoder architecture (rescae), which maintains a similar network structure to the UNet but removes skip connections to assess their contribution to model performance. All models were trained using identical hyperparameter configurations, including a learning rate of $1e-5$ (selected from candidates $1e-3, 1e-4, 1e-5$ as the optimal choice), 200 training epochs, and the AdamW optimiser with weight decay of $1e-4$. All models are evaluated on the ensemble test dataset of the Chimney fire event.

The results in Figure 1 demonstrate that the UNet Architecture with attention mechanisms generally outperforms the UNet basic architecture without attention blocks, though the improvement varies across metrics and dataset sizes. For instance, at training dataset size 500, the attention-enhanced UNet achieves superior performance in MSE ($5.60 \times 10^{-3}$ vs $6.00 \times 10^{-2}$), SSIM ($8.92 \times 10^{-1}$ vs $8.73 \times 10^{-1}$), and FID ($4.23 \times 10^{1}$ vs $6.16 \times 10^{1}$) whilst showing comparable performance in other metrics such as PSNR. However, at smaller dataset sizes like 100, the performance differences become less pronounced, with some metrics showing marginal improvements whilst others exhibit comparable or slightly inferior performance. In contrast, the comparison between UNet architectures and the residual AutoEncoder architecture reveals more substantial performance differences. The UNet structures consistently demonstrate significant improvements across multiple metrics, highlighting the importance of skip connections in preserving fine-grained information throughout the encoding-decoding process.

The related source code, scripts, data, and experimental results have been uploaded to Zenodo (`https://zenodo.org/records/15699653`) (Yu et al., 2025). The experimental results can be found in the `out` directory.

Following our previous response, it is important to clarify that the primary focus of this study is not to establish the superiority of various model architectures, but rather to investigate the advantages of diffusion-trained stochastic models over deterministic models in simulating wildfire uncertainty. This finding is further supported by our new numerical experiments: across all three neural network architectures, diffusion-based ensemble predictions (bar chart with slashes) substantially outperform their deterministic counterparts (bar chart without slashes) as shown in Figure 1.

**Our previous response (copied here for reference)**

We thank the reviewer for the detailed comments and suggestions on our manuscript. However, we believe that some key aspects of our work may have been overlooked.

1. The reviewer has repeatedly suggested that a benchmark comparing "DDPM vs. ConvLSTM (or other NNs)" is necessary for the manuscript. We would like to first clarify that sampling algorithms (e.g., DDPM or DDIM) and neural network architectures (e.g., ConvLSTM or U-Net) are fundamentally two different things. Sampling algorithms define how noise is added during the forward diffusion process and removed during the reverse (denoising) process (e.g., markovian in the case of DDPM and deterministic non-markovian in the case of DDIM, see Austin et al. (2021); Song et al. (2022), whereas different neural network architectures (such as U-Net or Transformer) could be chosen to train this denoising procedure. We believe the reviewer may be confused about this fundamental concept. We can not compare a sampling/denoising method against a neural network structure.

The main objective of our experiments here is to compare a diffusion-based generative training algorithm with the deterministic training method (based on MSE) for wildfire prediction,

rather than to evaluate different neural network architectures. Therefore, we compared the performance of a conditional diffusion model based on U-Net to that of the same U-Net trained using a deterministic approach. In addition, using a different neural network architecture might improve the accuracy of deterministic training, but it would not provide probabilistic predictions or capture the uncertainty of fire propagation. And also, the new network architecture will likely improve the diffusion model's performance as well. This does not qualitatively affect our comparison of diffusion and deterministic training.

We thank the reviewer for this question and will clarify the differences between neural network architectures and diffusion sampling algorithms for non-expert readers in ML.

2. Regarding the novelty of our work, although we agree that diffusion models have recently been applied in geoscience, to the authors' knowledge, this is the first study to apply diffusion-based generative AI to wildfire spread prediction (see a recent review paper by Xu et al. (2025)). In fact, to our knowledge, only one previous GMD publication (Elena Tomasi et al., April 2025) has applied a latent diffusion model to a downscaling task. Therefore, we believe that our paper is the first to use a conditional diffusion model for dynamical-system prediction in GMD.

More importantly, our diffusion model is trained using data generated from a stochastic simulator of wildfire. Therefore, we examine if the ensemble generated by the diffusion model could represent the stochasticity of the original physics model, which brings a unique contribution and insight to the community. We have also designed a specific validation procedure to compare the two ensembles generated by the stochastic physics model and the diffusion AI model, as described in Section 2.2.2 and illustrated in Figures 3 and 7 of our manuscript.

We believe that developing a surrogate model using diffusion-based generative method to capture uncertainties in stochastic physics simulators is novel within geoscience, if not in the broader computational physics field.

Following the reviewer's suggestion, we will perform additional hyperparameter tuning in the revised manuscript to improve our diffusion model's performance. However, as noted, our primary objective is to demonstrate a generative diffusion model's ability to capture the stochasticity of the physics-based model, which our current results already successfully achieve.

3. The reviewer repeatedly refers to DDPM as our denoising approach and points out its computational inefficiency. However, in our manuscript we employ the DDIM algorithm, as clearly stated in the first sentence of Section 3.1.4, in Equation 9 on page 14, and in Algorithm 2 on page 15. We also explain our choice of DDIM over DDPM, indeed specifically for its superior computational efficiency, in Section 3.1 on page 15. Thus, we believe the reviewer may have overlooked some important statements in our methodology section.

**Bibliography**

Austin, J., Johnson, D. D., Ho, J., Tarlow, D., and van den Berg, R.: Structured Denoising Diffusion Models in Discrete State-Spaces, in: Advances in Neural Information Processing Systems, vol. 34, pp. 17 981–17 993, Curran Associates, Inc., 2021.

Song, J., Meng, C., and Ermon, S.: Denoising Diffusion Implicit Models, https://doi.org/10.48550/arXiv.2010.02502, 2022.

Xu, Z., Li, J., Cheng, S., Rui, X., Zhao, Y., He, H., Guan, H., Sharma, A., Erxleben, M., Chang, R., et al.: Deep learning for wildfire risk prediction: Integrating remote sensing and

environmental data, ISPRS Journal of Photogrammetry and Remote Sensing, 227, 632–677, 2025.

Yu, W., Ghosh, A., Finn, T., Arcucci, R., Bocquet, M., and Cheng, S.: A Probabilistic Approach to Wildfire Spread Prediction Using a Denoising Diffusion Surrogate Model, https://doi.org/10.5281/zenodo.15669653, 2025.

---

## Author Comment (AC2)

**Reply to referee 2**

**0.1 General Comments:**

In this paper, a probabilistic method using a denoising diffusion surrogate model is applied to study the wildfire spread prediction, which has the advantage of quantifying the uncertainty. The study focuses on synthetic wildfire data generated by a probabilistic cellular automata-based simulator. The study is systematic, and the presentation of the results is detailed. I have a few minor suggestions, especially several clarification questions.

*We thank the Reviewer for the positive comments and the appreciation of our work. We have carefully addressed all the comments with additional numerical experiments, and we revised the manuscript accordingly. A point-to-point response is listed here below.*

**0.2 Specific Comments:**

1. The authors highlighted that "this study seeks to address the limitations of traditional deterministic wildfire forecasting methods." What about the existing stochastic or probabilistic models?

   *We agree with the Reviewer that stochastic models are widely used in wildfire modelling to capture extreme fire behaviour. In fact, the training and test data used in this work is generated via a physics-based stochastic cellular-automata fire simulator Alexandridis et al. (2008). Our goal here is not to compare conventional stochastic fire predictors with deep learning–based ones but rather to investigate whether a generative machine learning model can effectively simulate wildfire propagation dynamics by learning from and reproducing the stochastic behaviour of a physics-based CA model.*

   *Accordingly, we thoroughly compare the outputs of the proposed diffusion-based wildfire predictor against the original CA model and show that, with the diffusion approach, we can numerically represent the probability density function of the CA outputs. By contrast, conventional deterministic models typically predict only the mean of possible scenarios and therefore lose the ability to capture extreme fire events. Following the Reviewer's suggestion, we have highlighted this aspect in the introduction of the revised manuscript (page 4) "Evaluation uses data from a probabilistic cellular-automata emulator incorporating canopy cover, canopy density, and slope. We analyse the stochastic outputs to assess whether the diffusion model captures the physics model's uncertainties."*

2. The authors may add more explicit statements to highlight the novelty of this work. Is this just an application or are there existing improvements in the techniques?

   *This study is, to our knowledge, the first to apply diffusion-based generative AI to wildfire spread prediction (Xu et al., 2025). More importantly, our model is trained on data from a stochastic wildfire simulator, allowing us to test whether the diffusion model's ensemble reproduces the stochasticity of the original physics model. We designed a dedicated validation procedure to compare ensembles from both models, as detailed in Section 2.2.2 and shown in Figures 3 and 7. We believe that developing a surrogate model using diffusion-based generative methods to capture uncertainties in stochastic physics simulators is novel in computational science. Following the Reviewer's suggestion, we have added a paragraph in the introduction of the revised manuscript (page 1) to highlight it "To the authors' knowledge, no existing work has used diffusion-based generative models*

*to predict fire spread in the literature. Furthermore, we believe that employing such surrogates to capture uncertainties in stochastic physics simulators is novel in computational science."*

3. The interpretability of probabilistic forecasting needs more discussion. These forecasts indeed provide UQ. But is such a UQ reliable and accurate?

   We have thoroughly compared the estimated probability distribution of our generative model with that of the physics-based model to ensure the diffusion model provides accurate uncertainty quantification of possible fire-spread scenarios. Following the Reviewer's suggestion, we have highlighted the comparison between the ensemble diffusion model's UQ and the numerical UQ of the original physics-based CA model in the revised manuscript (page 22) "*In particular, the strong FID and KL results indicate that the diffusion model's estimated probability distribution is reliable, as it closely matches that of the original physics-based CA model.*".

4. The physical mechanism is quite complicated, and therefore several variables are involved in the models. How sensitive is the diffusion emulator with respect to the perturbation of each parameter/input?

   In fact, to generate different scenarios for the dataset, we randomised the initial parameters following our previous work (Cheng et al., 2022). Consequently, both the training and test datasets contain fire scenarios generated with different initial parameters. We have added a description in section 2.1 (page 7) of the revised manuscript to clarify this. "*The operational parameters $p_h$, $a$, $c_1$, and $c_2$ influence the fire forecast, where $a$ is the slope effect coefficient and $c_1$, $c_2$ are the wind effect coefficients. The detailed formulations of the slope and wind effects are described in Cheng et al. (2022)...Training data is generated via Latin Hypercube Sampling (LHS) within the range of an ensemble of perturbed parameter sets:*"

   $$p_h \in [0.00,\ 0.35],\ a \in [0.00,\ 0.14],\ c_1 \in [0.00,\ 0.12],\ c_2 \in [0.00,\ 0.40] \tag{1}$$

   *where the parameter ranges are based on the previous study by Cheng et al. (2022).*

   Following the Reviewer's question, we have also added an analysis in page 23 of the manuscript "*It is also worth mentioning that the CA model parameters $p_h$, $a$, $c_1$, and $c_2$ are randomly perturbed when generating the training and test datasets. The numerical results presented in Table 3 further demonstrate the robustness of the proposed diffusion model against variations in fire modelling parameters.*"

5. The role of some of the details of the emulator's components needs to be discussed. For example, what if the attention mechanism is removed?

   We thank the Reviewer for pointing out this question regarding the architectural components of our model. Following the Reviewer's suggestion, we have conducted an ablation study examining the role of attention mechanisms and other architectural details, which we present in the Appendix D: Ablation study on model architecture of our paper and illustrate in Figure D1 (page 32) of the revised manuscript, summarised here below.

[Figure]

Figure 1: Ablation study comparing model performance across different training dataset sizes and architectures.

Our ablation study evaluates three distinct neural network architectures across varying training dataset sizes {50, 100, 200, 500, and 900 samples}, comparing stochastic diffusion models against deterministic approaches, as shown in Figure D1. The architectures examined include: (1) a UNet architecture with attention mechanisms (unet_atten) as employed in our main study; (2) a UNet basic architecture (unet_basic), representing a simplified variant without attention blocks; and (3) a residual AutoEncoder architecture (rescae), which maintains a similar network structure to the UNet but removes skip connections to assess their contribution to model performance. All models were trained using identical hyperparameter configurations, including a learning rate of $1e-5$ (selected from candidates $1e-3, 1e-4, 1e-5$ as the optimal choice), 200 training epochs, and the AdamW optimiser with weight decay of $1e-4$. All models are evaluated on the ensemble test dataset of the Chimney fire event.

The related source code, scripts, data, and experimental results have been uploaded to Zenodo (`https://zenodo.org/records/15699653`) (Yu et al., 2025). The experimental results can be found in the `out` directory.

Turning to the specific question about attention mechanisms, our experimental results demonstrate that the attention-enhanced UNet generally outperforms the UNet basic architecture without attention blocks, though the benefits vary across metrics and dataset sizes, as shown in Figure 1. It is worth noting that at smaller dataset sizes (e.g., 100 samples), the performance differences between architectures with and without attention become less pronounced. This observation may partially be attributed to the specific characteristics of our experimental data, including its moderate spatial resolution ($64 \times 64$), which might not fully utilise the representational capabilities of attention mechanisms. We suggest that in scenarios involving more complex spatial patterns or higher-resolution data, the benefits of attention mechanisms could potentially be more pronounced.

Our ablation study demonstrates that whilst architectural improvements (such as attention mechanisms) provide meaningful enhancements, the more substantial performance gains arise from the fundamental shift from deterministic to stochastic modelling approaches. The consistent superiority of diffusion models across all architectural variants reinforces our central ideas that stochastic models are inherently better suited for capturing the uncertainty and variability characteristic of wildfire behaviour.

6. Some details about the background should be added. For example, subsampling frames at 20-hour intervals is used for training. Why is such a specific number chosen? There are a lot of mathematical details, but some of the physics or reasoning are missing.

We thank the Reviewer for pointing this out. The 20-hour interval was chosen to ensure sufficiently large time steps for observing meaningful differences across fire-spread stages. This time interval is consistent with recent research works (Kondylatos et al., 2022; Huot et al., 2022). Following the Reviewer's suggestion, we have added a paragraph in the revised manuscript (page 9) to explain the reason of this choice "*To enlarge the prediction window and maintain substantial differences between successive fire states, frames are subsampled from each simulation at intervals of 10 time steps (*20 *hours), yielding six frames per wildfire event.*"

**Bibliography**

Alexandridis, A., Vakalis, D., Siettos, C. I., and Bafas, G. V.: A Cellular Automata Model for Forest Fire Spread Prediction: The Case of the Wildfire That Swept through Spetses Island in 1990, Applied Mathematics and Computation, 204, 191–201, https://doi.org/10.1016/j.amc.2008.06.046, 2008.

Cheng, S., Jin, Y., Harrison, S. P., Quilodrán-Casas, C., Prentice, I. C., Guo, Y.-K., and Arcucci, R.: Parameter Flexible Wildfire Prediction Using Machine Learning Techniques: Forward and Inverse Modelling, Remote Sensing, 14, 3228, https://doi.org/10.3390/rs14133228, 2022.

Huot, F., Hu, R. L., Goyal, N., Sankar, T., Ihme, M., and Chen, Y.-F.: Next day wildfire spread: A machine learning dataset to predict wildfire spreading from remote-sensing data, IEEE Transactions on Geoscience and Remote Sensing, 60, 1–13, 2022.

Kondylatos, S., Prapas, I., Ronco, M., Papoutsis, I., Camps-Valls, G., Piles, M., Fernández-Torres, M.-Á., and Carvalhais, N.: Wildfire danger prediction and understanding with deep learning, Geophysical Research Letters, 49, e2022GL099 368, 2022.

Xu, Z., Li, J., Cheng, S., Rui, X., Zhao, Y., He, H., Guan, H., Sharma, A., Erxleben, M., Chang, R., et al.: Deep learning for wildfire risk prediction: Integrating remote sensing and environmental data, ISPRS Journal of Photogrammetry and Remote Sensing, 227, 632–677, 2025.

Yu, W., Ghosh, A., Finn, T., Arcucci, R., Bocquet, M., and Cheng, S.: A Probabilistic Approach to Wildfire Spread Prediction Using a Denoising Diffusion Surrogate Model, https://doi.org/10.5281/zenodo.15699653, 2025.

---

## Author Response (AR2)

**Reply to referee 1**

**General Comments:**

This study proposes a probabilistic surrogate model for wildfire spread prediction using a denoising diffusion probabilistic model. The authors have revised the manuscript based on the previous review comments. In particular, the addition of the ablation study is a valuable improvement that enhances the overall contribution of the paper. In my previous review, I provided five main comments regarding (1) novelty, (2) scientific discussion to enhance the paper's academic value, (3) justification, (4) ablation study, and (5) insights specific to wildfire prediction. Among these, point (4) has been addressed through the addition of an ablation study in the Appendix. Also, I appreciate the authors' effort to better explain the distinction between diffusion-based sampling algorithms and neural network architectures, as well as their clarification that DDIM, not DDPM, was used in their study. While these clarifications are helpful, I still find that several of my core concerns regarding scientific novelty, comparative justification, and domain-specific insights remain only partially addressed.

We thank the Reviewer for the constructive follow-up. In this revision we have (i) expanded the Introduction (page 3 of the manuscript) to articulate the methodological novelty of the diffusion surrogate relative to GAN-, VAE- and flow-based surrogates and to explain why diffusion is well suited to probabilistic wildfire spread prediction; (ii) clarified the scientific and domain-specific insights by discussing how the diffusion ensemble captures stochastic behaviour and uncertainty structures (pages 3–4 of the manuscript); (iii) strengthened the justification for using the calibrated and validated CA simulator, detailing its grounding in real geophysical inputs and prior validation (pages 6–8 of the manuscript); (iv) retained and highlighted the new ablation study in the Appendix D of the manuscript. In addition, we added discussion on the computational efficiency comparison between the CA model and the diffusion surrogate and a clearer explanation of the probabilistic metrics employed, in particular the Hit Rate. We hope these changes address the Reviewer's remaining concerns on novelty, comparative justification, and domain-specific insight.

**Specific Comments:**

**1. On novelty and scientific contribution:**

I appreciate the authors' clarification that this may be the first GMD submission applying diffusion-based generative AI to wildfire spread. However, in my opinion, novelty in journal publications should not be judged solely by first application, but rather by the new methodological or domain-specific in- sights it provides. While the study's focus on capturing stochasticity from a physics-based simulator is interesting, this concept aligns closely with a broader body of surrogate model- ing literature using GANs, VAEs, and normalizing flows. To strengthen the contribution, I would encourage the authors to discuss how their diffusion-based surrogate differs, in principle and in outcome, from these earlier probabilistic surrogate approaches.

We appreciate the Reviewer for this insightful remark on novelty and scientific contribution. In the revised manuscript, we have therefore expanded the discussion in the Introduction (page 3) to more clearly position our work within the broader literature on probabilistic surrogate modelling with GANs, VAEs and normalising flows, and to explain how our diffusion-based surrogate differs in principle and outcome. In particular, we now summarise known limitations

of these earlier approaches for high-dimensional, uncertainty-aware emulation. In summary, as shown in our previous work (Cheng et al., 2023), VAEs can generate realistic fire scenarios at a given time step; however, their bottleneck architecture limits their ability to produce conditional predictions of fire spread. We highlight the advantages of diffusion models as likelihood-based, stably trained generative models with strong sample quality and mode coverage, and point to recent geoscientific applications where diffusion models have been successfully used as probabilistic surrogates and ensemble generators.

We also clarify that the novelty of our contribution is not limited to being one of the first applications of diffusion models in GMD. First, to the best of our knowledge, this study is the first to apply a diffusion model as a predictive surrogate for wildfire spread. Second, and more importantly from a methodological perspective, our study uses diffusion modelling to learn a stochastic dynamical system where the full conditional distribution of the reference simulator is known. This enables a direct comparison between the uncertainty produced by the diffusion surrogate and that of the underlying stochastic model, providing a quantitative assessment of whether the generative model recovers the true stochastic behaviour. We regard this controlled, distribution-level evaluation as an important contribution of the work in particular within the field of geoscientific surrogate modelling.

**2. Experimental Setting, Discussion and Insight:**

General Comment: The authors highlight that the diffusion model ensemble can reproduce the stochasticity of the physics simulator. This is indeed a valuable result, but the paper could further benefit from a more explicit discussion of what this implies scientifically, for example, whether the generative model reveals any new understanding about wildfire spread variability, sensitivity to environmental conditions, or uncertainty structures. Such insights would elevate the manuscript beyond a technical demonstration toward a contribution of broader geoscientific relevance.

1. As I understand it, the manuscript uses a cellular automaton (CA) model to generate the training data. Although the study utilizes the Chimney Fire (2016) and Ferguson Fire (2018) cases, it appears that only topographic and vegetation data are used, and that no actual wildfire observation data are incorporated. In addition, please clarify whether the CA model used for training the diffusion model was validated against any real wildfire observation data prior to its use.

   We thank the reviewer for highlighting the role of real wildfire observations in the context of training data. In the revised manuscript (pages 6 to 8), we clarify that the diffusion model is intentionally trained and validated on data generated by a calibrated and validated cellular automaton (CA) wildfire simulator, rather than on direct perimeter or burn-scar observations. The goal of this study is to develop a proof-of-concept generative surrogate for a stochastic wildfire-spread model, where the CA simulator defines the target conditional distribution that the diffusion model aims to learn.

   To address the concern regarding validation, we emphasise in the revised CA section that the specific CA formulation used here is directly based on previously published work (Alexandridis et al., 2008; Cheng et al., 2022), in which the model parameters were systematically calibrated and its behaviour validated against observed wildfire-spread patterns. Additional literature demonstrates that CA-based wildfire simulators are capable of reproducing key aspects of real fire propagation across diverse environments when appropriately parameterised. For example, Freire and DaCamara (2019) validated their CA model using the 2012 Algarve wildfire in southern Portugal, and Trucchia et al. (2020)

demonstrated the operational PROPAGATOR system across several Mediterranean fire events in Italy and Spain.

Although the present study does not incorporate continuous time-resolved wildfire observations for training, we clarify that the CA simulations are driven by real geophysical inputs from the Chimney Fire (2016) and Ferguson Fire (2018), including canopy cover, canopy density, topographic slope and local wind, ensuring that the generated ensembles remain anchored in realistic environmental forcing. The revised manuscript (page 4) explicitly explains this design choice and the scientific motivation behind using a controlled, validated stochastic simulator for initial surrogate development.

2. I understand that CA models are widely used in the wildfire modeling community due to their simplicity and computational efficiency. However, compared to physics-based models such as WRF-Fire, I guess their ability to reproduce real wildfire behavior would be generally limited. Therefore, I would appreciate clarification on whether the CA model used in this study has been calibrated or validated to ensure a reasonable level of physical realism. In addition, please include a brief discussion on what challenges might be encountered if a similar diffusion-based surrogate modeling framework were to be developed using a physics-based model such as WRF-Fire—for example, in terms of data availability and computational cost.

We thank the reviewer for this important comment. In addition to the clarifications already added to the manuscript (pages 6–8), we have now further strengthened the discussion by conducting a new computational-efficiency comparison between the calibrated CA model used in this study and inference using our diffusion surrogate (DDIM sampling). This experiment quantifies the practical cost of producing large stochastic wildfire ensembles, which would be required if a physics-based model such as WRF-Fire were used instead.

As highlighted in the revised manuscript, the CA simulator employed here is directly based on a validated CA framework (Alexandridis et al., 2008). That previous study presented parameter calibration against observed wildfire spread, demonstrating that the model holds the capability of reproducing key aspects of real-world behaviour when driven by realistic environmental forcing. Additional literature further supports the capacity of CA-based models to emulate essential wildfire-spread dynamics across diverse landscapes. For example, Freire and DaCamara (2019) applied a CA framework to the large 2012 Algarve wildfire in Portugal—covering approximately 25,000 ha—and showed that the simulator successfully reproduced the temporal evolution and spatial extent of the burned area, thereby confirming its practical utility for fire-behaviour analysis. Similarly, Trucchia et al. (2020) created the operational PROPAGATOR CA wildfire simulator and demonstrated its ability to simulate several Mediterranean fire events in Italy and Spain in a short amount of time. These studies indicate how efficiency and physical interpretability of CA-based semi-physical approaches can be in real-world forecasting situations.

To clarify the computational implications raised by the reviewer, we performed an additional benchmark comparing the wall-clock inference time (using an NVIDIA RTX 6000 GPU and 96 CPU cores from an Intel(R) Xeon(R) Gold 6442Y processor) of the semi-physical CA model and the diffusion surrogate using DDIM sampling, tested under several batch sizes.

| Method | Batch Size | Total Samples | Total Time (s) | Avg. Sample Time (ms) | Throughput (samples/s) |
|---|---|---|---|---|---|
| Diffusion | 1 | 400 | 103.17 | 257.91 | 3.88 |
| Diffusion | 10 | 400 | 34.56 | 86.40 | 11.57 |
| Diffusion | 20 | 400 | 34.72 | 86.79 | 11.52 |
| Diffusion | 50 | 400 | 38.53 | 96.33 | 10.38 |
| Diffusion | 100 | 400 | 39.75 | 99.38 | 10.06 |
| CA Model | N/A | 50 | 177.81 | 3556.20 | 0.28 |

Table 1: Comparison of computational efficiency between the diffusion surrogate (DDIM inference) and the CA simulator. Throughput is measured as samples per second.

[Figure]

Figure 1: Runtime comparison.

The comparison—also summarised visually in Figure 1—shows that:

- The CA model requires  3.56 seconds per simulation, while
- The diffusion surrogate requires 0.086–0.26 seconds per simulation, depending on batch size.

Thus, diffusion is 14–40× faster per sample and up to  50× higher throughput.

It is also important to emphasise that fully coupled physics-based wildfire simulators—such as WRF-Fire (Mandel et al., 2011), FIRETEC (Linn and Cunningham, 2005) models—are significantly more computationally demanding than the CA framework considered here. Such models typically require hours to days of wall-clock time on HPC systems for a single simulation because they explicitly resolve fire–atmosphere interactions, turbulent flows, radiative transfer and detailed fuel processes. Generating the tens of thousands of realisations required to train a generative model would therefore be computationally difficult with these physics-based systems.

Against this backdrop, machine learning surrogates such as diffusion models offer a clear practical advantage: after training, they can generate large ensembles at a fraction of the computational cost required by physics-based simulators. The speed-up observed relative to the CA model therefore represents a conservative lower bound, and the efficiency benefit of diffusion-based emulators would be even greater when replacing substantially more expensive physical models.

3. If the CA-based wildfire simulations themselves have limited physical realism, it is unclear how much scientific value an AI model that merely surrogates such simulations can offer. In meteorology, although each phenomenon occurs only once, probabilistic forecasting

has advanced through approaches such as optimizing models with the Continuous Ranked Probability Score (CRPS) and evaluating forecast distributions against single realizations. Would it be possible to adopt a similar approach for probabilistic wildfire prediction, using real fire observations rather than CA-generated data? I understand that, unlike in meteorology, it may be difficult to directly apply such approaches due to the lack of consistent and continuous wildfire observations. However, please consider adding a discussion on whether more physically realistic surrogates could be developed in the future by leveraging physics-based models or real observation data.

We thank the reviewer for raising this important point. We emphasise that the CA-based wildfire simulations used in this study are not arbitrary or purely heuristic: they are generated by a calibrated and previously validated semi-physical model (Alexandridis et al., 2008; Cheng et al., 2022), driven by real geophysical inputs from the Chimney and Ferguson Fire cases. Our previous research and supporting literature indicate that such cellular automata formulations can accurately replicate essential spatiotemporal features of wildfire propagation when appropriately parameterized. Consequently, they offer a sufficiently realistic and scientifically significant testbed for the proof-of-concept aim of this study.

Regarding the reviewer's suggestion on adopting CRPS evaluation, we note that such methods require only a single realised trajectory, which makes them attractive for meteorological forecasting. In principle, a diffusion surrogate can also be trained and evaluated against real wildfire observations, and extending our framework in that direction will be an important step for future work. In this study, however, we intentionally rely on CA-generated data because the simulator provides access to the entire stochastic distribution of fire-spread outcomes. This enables a form of validation that is not possible with observational data, as it allows us to compare the ensemble produced by the diffusion model with the true reference distribution rather than with a single realised perimeter. The ability to perform this distribution-level verification is therefore a contribution of our work and a key reason for choosing a controlled CA-based setting for the initial development of the surrogate.

In summary, a central contribution of our manuscript is precisely this: we design and evaluate a diffusion surrogate in a controlled, physically grounded environment where the target stochastic dynamics are known. This controlled validation strategy allows us to rigorously assess whether the generative model can reproduce the underlying conditional distribution—an assessment that would be impossible using sparse and inconsistently sampled real wildfire observations. We have now explicitly highlighted this rationale and contribution in the revised manuscript (page 4).

At the same time, we agree that future surrogates should move toward more physically comprehensive and observation-informed datasets. Accordingly, we have strengthened the Future Work (page 28) section to outline next steps.

**3. Comparison with other AI-based approaches:**

I acknowledge the authors' point that DDPM/DDIM is a sampling algorithm rather than a neural architecture, and that ConvLSTM represents a distinct modeling approach. However, my original comment was not intended to suggest a direct algorithmic comparison between DDPM and ConvLSTM, but rather to emphasize the need for a comparative baseline that reflects widely used probabilistic or spatiotemporal modeling strategies in wildfire research. Comparing a diffusion-based model only with its deterministic counterpart (UNet with MSE loss) would not provide a sufficient justification for adopting the diffusion framework. The Introduction provides a solid overview of existing AI-based wildfire prediction models, including

UNet, CNN-LSTM, and Transformer architectures (e.g., Shah and Pantoja, 2023; Chen et al., 2024b; Khennou and Akhloufi, 2023). Given this context, it would be helpful for the authors to clarify why the UNet architecture was chosen as the baseline in this study and to discuss its suitability in relation to the previously reviewed AI-based wildfire prediction models.

We appreciate the Reviewer's comment and note that we have already addressed the architectural question in our earlier response and Appendix D in the revised manuscript. Here we briefly reiterate the main points for clarity.

In this study, we adopt a UNet-type encoder–decoder as the backbone architecture for both the deterministic and diffusion-based models. As discussed in our previous response, UNet remains one of the most widely used and empirically validated architectures for spatial prediction tasks in wildfire modelling and remote sensing. Our earlier work (Zhou et al., 2025) demonstrated that UNet provides strong predictive skill, stable optimisation behaviour, and favourable interpretability compared with alternative architectures such as Swin-Transformer models. Its multi-scale feature extraction and skip-connection structure make it a robust and sufficiently expressive foundation for diffusion-based generative modelling.

We would also like to emphasise that UNet and its attention-enhanced variants constitute the standard and most widely adopted backbones for diffusion models. The seminal DDPM paper (Ho et al., 2020), the improved classifier-free and attention-enhanced diffusion models of Dhariwal and Nichol (2021), and recent large-scale latent diffusion frameworks (Rombach et al., 2022) all employ UNet architectures. Their consistent success across domains justifies our choice of UNet as the baseline architecture for a fair and interpretable comparison between deterministic and diffusion-based training paradigms.

[Figure]

Figure 2: Ablation study comparing model performance across different training dataset sizes and architectures.

As summarised in the earlier response letter, we also conducted an ablation study (Figure 2) comparing three architectures: an attention-enhanced UNet, a basic UNet, and a residual autoencoder (ResCAE). All models were trained under identical settings. While attention mechanisms yield modest improvements, our results show that the diffusion-based formulation consistently outperforms deterministic counterparts across all architectures. Moreover, the ResCAE backbone exhibits unstable convergence when used within the diffusion framework, reinforcing UNet as the most reliable and practically effective choice for our study.

To avoid redundancy, we refer the Reviewer to the detailed explanation and experimental evidence already included in the revised manuscript (page 32) and previous response. Here we simply emphasise that the purpose of this work is not to benchmark neural architectures, but to demonstrate the methodological benefits of adopting a diffusion-based probabilistic framework. Architectural differences are secondary to this shift, and UNet provides a strong, well-established baseline for a fair and interpretable comparison.

**Reply to referee 2**

**General Comments:**

This manuscript introduces a probabilistic wildfire spread surrogate model based on conditional denoising diffusion and compares it against a deterministic surrogate with an identical architecture, trained on synthetic data generated by a probabilistic CA fire spread simulator. The topic is timely, the motivation is well-grounded, and the methodological implementation appears sound. The comparative experimental design is clean, as the two models share the same backbone and differ solely in training paradigm, enabling a fair attribution of the performance gain to the diffusion-based stochastic formulation rather than architectural complexity. The manuscript is in general clearly written and the results are convincing.

We thank the reviewer for the positive overall assessment of our work and for the detailed feedback. In revising the manuscript, we have substantially clarified the study's scientific contribution and strengthened the justification for our methodological choices. We now explicitly motivate the use of a calibrated and previously validated CA model as a controlled testbed (pages 6–8), explain how this setting enables rigorous comparison between the diffusion surrogate and the reference stochastic dynamics (page 4), and outline how future work will extend the framework to more physically comprehensive and observation-informed datasets (page 28). In addition, we clarify that our evaluation compares the probability density functions produced by the CA simulator and the diffusion model, both yielding continuous values. Because the Brier Score (Redelmeier et al., 1991) is designed for binary observed outcomes, it is not well suited to this setting; the Hit Rate (HR) therefore serves as a more appropriate measure for assessing whether the diffusion model reproduces the variability encoded in the reference stochastic simulator. Furthermore, we have added a quantitative runtime comparison in this letter to make the computational advantages of the diffusion surrogate explicit. We hope that these revisions adequately address the reviewer's concerns and further strengthen the clarity and scientific value of the manuscript.

**Specific Comments:**

1. First, the abstract is currently too verbose and narrative in style; it should be condensed, and — more importantly — it should report not only relative gains (e.g. percentage reduction in MSE or FID) but also the corresponding absolute levels of accuracy, such that the reader can judge the practical quality of predictions rather than merely their improvement over a baseline.

   We thank the Reviewer for this helpful observation. In the revised manuscript, we have condensed the abstract to focus on the core methodological contribution. While the Reviewer suggested providing absolute accuracy levels in the abstract, we note that the dataset used in this study is not a standardised benchmark such as ImageNet(Deng et al., 2009) or WeatherBench(Rasp et al., 2020). For this reason, isolated absolute accuracy values may have limited scientific interpretability for readers who are unfamiliar with the experimental setup. To ensure clarity and avoid potential misinterpretation, we therefore report the absolute accuracy values in the Introduction as suggested by the reviewer.

   Following this approach, the resubmitted manuscript now states in the final paragraph of the Introduction that, the diffusion-based surrogate achieves an MSE of 0.0067 versus 0.0218 and an FID of 49.0 versus 165.9 relative to the deterministic baseline. We also provide the relative improvements to assist interpretation. These additions allow readers

to form an informed view of the practical accuracy of the method once sufficient context has been provided. We further clarify in the abstract that both models share an identical network architecture, which ensures the transparency of the comparative setting. We hope that these revisions adequately address the Reviewer's concern.

2. Second, the manuscript implicitly treats CA ensemble outputs as the reference truth for training and validation. This design choice is understandable for proof-of-concept surrogate development, but it is not explicitly justified. The authors should clearly state whether the goal is (i) to build a surrogate for the CA model itself, in which case CA defines the target distribution by construction, or (ii) to approximate physical wildfire evolution ultimately driven by real events, in which case the absence of any experiment with real observational data (e.g. perimeter burn scar maps) requires justification. Articulating this will help avoid the impression that the model only learns the statistics of a simulator rather than reality.

We thank the reviewer for the helpful comment. We clarify that the primary objective of this study is to demonstrate that diffusion models are capable of learning and reproducing the stochastic dynamics of wildfire spread. To rigorously test this capability, the surrogate must be evaluated against a reference model that exhibits meaningful physical behaviour and realistic variability.

For this proof-of-concept stage, we therefore rely on a calibrated and previously validated CA wildfire simulator (Alexandridis et al., 2008; Cheng et al., 2022). Although less complex than fully coupled physics-based models, the CA framework provides a computationally efficient and physically grounded way to capture key aspects of real wildfire behaviour—such as anisotropic spread, wind-driven acceleration, and topography-dependent propagation—while producing stochastic ensembles necessary for evaluating generative models. In this sense, the CA model offers a suitable proxy for the complexity of real fires while allowing controlled, repeatable experimentation. We have now explicitly stated in the manuscript that designing and validating the diffusion surrogate within this controlled CA-based setting is itself an important contribution (has been mentioned in page 4 of the revised manuscript), as it provides a clear and verifiable benchmark for assessing whether the diffusion model can recover a known stochastic target distribution.

3. Third, the stated advantage over CA-based ensemble simulation should be made explicit. If the improvement is primarily computational (i.e. the diffusion surrogate approximates CA but with orders-of-magnitude lower cost per ensemble realisation), this should be quantified. If, instead, the diffusion surrogate contributes additional information beyond CA (e.g. sharper spatial morphology, regularisation of small-scale artefacts, or stability when uncertainty is high), this claim should be discussed and supported; otherwise the benefit over simply running CA multiple times may appear unclear.

We thank the reviewer for this important comment. We now explicitly state that the primary advantage of the diffusion surrogate over repeated CA ensemble simulations is computational efficiency, and we have added a new quantitative benchmark to the revised manuscript (Table 1 and Figure 1). These results show that the calibrated CA model requires approximately 3.56 seconds per simulation, whereas DDIM sampling with the diffusion surrogate requires only 0.086–0.26 seconds per simulation, depending on batch size. This corresponds to a 14–40× reduction in per-sample runtime and up to a 50× increase in throughput, demonstrating that the diffusion model can produce large stochastic ensembles orders of magnitude more efficiently than the CA simulator.

Beyond computational savings, we have clarified in the text (pages 6 to 8) that the diffusion surrogate does not aim to exceed the CA simulator in physical realism; rather, the goal of this proof-of-concept study is to show that diffusion models can faithfully reproduce complex stochastic fire-spread dynamics generated by a validated semi-physical

simulator. This controlled evaluation—now explicitly described as one of the contributions of the manuscript.

To avoid overstating claims, we do not assert that the diffusion surrogate provides additional physical information beyond the CA model at this stage. Instead, the revised manuscript clearly positions the surrogate as a computationally efficient and distribution-preserving emulator of a validated stochastic simulator. As noted in the expanded Future Work section, future efforts will focus on incorporating richer environmental forcings and assessing performance against real world data, which will allow us to explore potential benefits beyond emulation of CA dynamics.

4. Finally, since the core methodological claim is probabilistic, the evaluation would be strengthened by including at least one standard uncertainty-aware verification measure such as Brier score, reliability diagrams, or sharpness–reliability diagnostics. This would complement the present set of similarity metrics and substantiate the claim that the method not only generates ensembles but also produces statistically meaningful probability forecasts.

Thank you for this constructive suggestion. We would like to clarify that the manuscript already incorporates a probabilistic evaluation through the Hit Rate (HR), which may not have been fully apparent in the original submission. To make this explicit, we have revised Appendix B (page 30) to contrast the HR with standard uncertainty-aware verification measures such as the Brier Score.

The Brier Score assesses probabilistic accuracy for binary events and is defined as

$$\text{BS} = \frac{1}{N} \sum_{i=1}^{N} (f_i - o_i)^2, \tag{1}$$

where $f_i$ is the forecast probability and $o_i \in \{0, 1\}$ is the observed binary outcome(Redelmeier et al., 1991). Because this metric requires observed events to be dichotomous, it is not directly applicable in our setting, where we would like to compare the probability distribution of the CA-generated targets and the diffusion-model outputs rather than binary occurrences.

For this reason, we introduce the Hit Rate (HR) as a tailored uncertainty-aware measure for continuous stochastic fields, defined in the manuscript as

$$\text{HR} = \frac{\sum_{\text{valid } i} \mathbf{1}\left(|\widehat{x}_i - x_i| < \epsilon\right)}{N_{\text{valid}}}, \tag{2}$$

where $\epsilon$ is the threshold that defines the acceptable difference between $\widehat{\mathbf{x}}_i$ and $\mathbf{x}_i$; $\mathbf{1}(\cdot)$ is the indicator function, which equals 1 when the condition inside it is true and 0 otherwise; and $N_{\text{valid}}$ is the number of valid target elements where $\mathbf{x}_i > 0$.

This metric quantifies the proportion of ensemble predictions that fall within an acceptable tolerance of the stochastic reference distribution. The HR therefore evaluates whether the diffusion model reproduces the intrinsic variability encoded in the CA simulator, fulfilling a role analogous to the Brier Score but appropriate for continuous fire-spread dynamics.

To clarify this discrepancy, we have expanded the explanation in Appendix B to explicitly justify the design of HR for probabilistic evaluation and to highlight its relevance in this continuous, simulator-based context.

**Bibliography**

Alexandridis, A., Vakalis, D., Siettos, C. I., and Bafas, G. V.: A Cellular Automata Model for Forest Fire Spread Prediction: The Case of the Wildfire That Swept through Spetses Island in 1990, Applied Mathematics and Computation, 204, 191–201, https://doi.org/10.1016/j.amc.2008.06.046, 2008.

Cheng, S., Jin, Y., Harrison, S. P., Quilodrán-Casas, C., Prentice, I. C., Guo, Y.-K., and Arcucci, R.: Parameter Flexible Wildfire Prediction Using Machine Learning Techniques: Forward and Inverse Modelling, Remote Sensing, 14, 3228, https://doi.org/10.3390/rs14133228, 2022.

Cheng, S., Guo, Y., and Arcucci, R.: A Generative Model for Surrogates of Spatial-Temporal Wildfire Nowcasting, IEEE Transactions on Emerging Topics in Computational Intelligence, 7, 1420–1430, https://doi.org/10.1109/TETCI.2023.3298535, 2023.

Deng, J., Dong, W., Socher, R., Li, L.-J., Li, K., and Fei-Fei, L.: ImageNet: A Large-Scale Hierarchical Image Database, in: 2009 IEEE Conference on Computer Vision and Pattern Recognition, pp. 248–255, ISSN 1063-6919, https://doi.org/10.1109/CVPR.2009.5206848, 2009.

Dhariwal, P. and Nichol, A. Q.: Diffusion Models Beat GANs on Image Synthesis, in: Advances in Neural Information Processing Systems, 2021.

Freire, J. G. and DaCamara, C. C.: Using Cellular Automata to Simulate Wildfire Propagation and to Assist in Fire Management, Natural Hazards and Earth System Sciences, 19, 169–179, https://doi.org/10.5194/nhess-19-169-2019, 2019.

Ho, J., Jain, A., and Abbeel, P.: Denoising Diffusion Probabilistic Models, https://doi.org/10.48550/arXiv.2006.11239, 2020.

Linn, R. R. and Cunningham, P.: Numerical Simulations of Grass Fires Using a Coupled Atmosphere–Fire Model: Basic Fire Behavior and Dependence on Wind Speed, Journal of Geophysical Research: Atmospheres, 110, https://doi.org/10.1029/2004JD005597, 2005.

Mandel, J., Beezley, J. D., and Kochanski, A. K.: Coupled Atmosphere-Wildland Fire Modeling with WRF 3.3 and SFIRE 2011, Geoscientific Model Development, 4, 591–610, https://doi.org/10.5194/gmd-4-591-2011, 2011.

Rasp, S., Dueben, P. D., Scher, S., Weyn, J. A., Mouatadid, S., and Thuerey, N.: WeatherBench: A Benchmark Dataset for Data-Driven Weather Forecasting, Journal of Advances in Modeling Earth Systems, 12, e2020MS002 203, https://doi.org/10.1029/2020MS002203, 2020.

Redelmeier, D. A., Bloch, D. A., and Hickam, D. H.: Assessing Predictive Accuracy: How to Compare Brier Scores, Journal of Clinical Epidemiology, 44, 1141–1146, https://doi.org/10.1016/0895-4356(91)90146-Z, 1991.

Rombach, R., Blattmann, A., Lorenz, D., Esser, P., and Ommer, B.: High-Resolution Image Synthesis with Latent Diffusion Models, https://doi.org/10.48550/arXiv.2112.10752, 2022.

Trucchia, A., D'Andrea, M., Baghino, F., Fiorucci, P., Ferraris, L., Negro, D., Gollini, A., and Severino, M.: PROPAGATOR: An Operational Cellular-Automata Based Wildfire Simulator, Fire, 3, 26, https://doi.org/10.3390/fire3030026, 2020.

Zhou, Y., Kong, R., Xu, Z., Xu, L., and Cheng, S.: Comparative and Interpretative Analysis of CNN and Transformer Models in Predicting Wildfire Spread Using Remote Sensing Data, Journal of Geophysical Research: Machine Learning and Computation, 2, e2024JH000 409, https://doi.org/10.1029/2024JH000409, 2025.